# Nonlinearities between inhibition and T-type calcium channel activity bidirectionally regulate thalamic oscillations

**Adam C Lu[1]\*, Christine Kyuyoung Lee[2], Max Kleiman-Weiner[3], Brian Truong[1], Megan Wang[4], John R Huguenard[5]\*, Mark P Beenhakker[1]\***

[1]Department of Pharmacology, University of Virginia, Charlottesville, United States; [2]Department of Neurosurgery, Massachusetts General Hospital, Boston, United States; [3]Department of Psychology, Harvard University, Cambridge, United States; [4]Princeton Neuroscience Institute, Princeton University, Princeton, United States; [5]Department of Neurology, Stanford University, Palo Alto, United States

**Abstract** Absence seizures result from 3 to 5 Hz generalized thalamocortical oscillations that depend on highly regulated inhibitory neurotransmission in the thalamus. Efficient reuptake of the inhibitory neurotransmitter GABA is essential, and reuptake failure worsens human seizures. Here, we show that blocking GABA transporters (GATs) in acute rat brain slices containing key parts of the thalamocortical seizure network modulates epileptiform activity. As expected, we found that blocking either GAT1 or GAT3 prolonged oscillations. However, blocking both GATs unexpectedly suppressed oscillations. Integrating experimental observations into single-neuron and network-level computational models shows how a non-linear dependence of T-type calcium channel gating on $GABA_B$ receptor activity regulates network oscillations. Receptor activity that is either too brief or too protracted fails to sufficiently open T-type channels necessary for sustaining oscillations. Only within a narrow range does prolonging $GABA_B$ receptor activity promote channel opening and intensify oscillations. These results have implications for therapeutics that modulate inhibition kinetics.

**\*For correspondence:**
al4ng@virginia.edu (ACL);
John.Huguenard@stanford.edu (JRH);
mpb5y@virginia.edu (MPB)

## Introduction

Neural circuits rely on a combination of intrinsic cellular properties and synaptic connections to generate large-scale electrical oscillations that drive behavior (*Getting, 1989*; *Marder and Calabrese, 1996*; *Nusbaum and Beenhakker, 2002*; *Huguenard and McCormick, 2007*). Following membrane hyperpolarization, such as that produced by synaptic inhibition, cortically projecting neurons of the thalamus [i.e. thalamocortical (TC) neurons] produce brief bursts of action potentials (*Llinás and Jahnsen, 1982*), a cellular property that maintains both sleep-related and seizure-related oscillations (*McCormick and Contreras, 2001*; *Huguenard and McCormick, 2007*; *Beenhakker and Huguenard, 2009*). Several studies have shown that CaV3.1 T-type calcium channels (T channels) sustain post-inhibitory rebound bursts in thalamocortical neurons by producing a relatively prolonged calcium-dependent, low-threshold spike (*Kim et al., 2001*; *Kim et al., 2003*; *Porcello et al., 2003*). These channels require membrane depolarization to open and hyperpolarization to recover (*Coulter et al., 1989*). Hyperpolarization-dependent recovery involves the removal of T channel inactivation (i.e. *de-inactivation*). As T channels are largely inactivated at resting membrane potentials, membrane hyperpolarization is necessary for robust rebound bursting (*Llinás and Jahnsen, 1982*; *Coulter et al., 1989*). While controlled voltage-clamp experiments have informed our understanding

of how neuronal membrane potential dynamics can affect T channel opening (*Gutierrez et al., 2001*), we still know little regarding channel behavior during physiological forms of synaptic inhibition when the underlying conductance rises and falls nonlinearly.

Reticular thalamic (RT) neurons serve as the main source of inhibitory, GABAergic input to thalamocortical neurons, especially in rodents (*Shosaku, 1985*; *Pinault and Deschênes, 1998*). Thalamocortical neurons express synaptic $\alpha_1\beta_2\gamma_2$ GABA$_A$ receptors, and two types of extrasynaptic receptors: GABA$_A$ ($\alpha_4\beta_2\delta$) and GABA$_B$ (*Pirker et al., 2000*; *Kulik et al., 2002*; *Jia et al., 2005*). Studies have shown that modulation of *synaptic* GABA$_A$ receptors on TC neurons has little effect on thalamocortical oscillations (*Sohal et al., 2003*; *Rovó et al., 2014*). In contrast, *extrasynaptic* receptors have been implicated in thalamocortical seizures, both for GABA$_A$ (*Cope et al., 2009*) and GABA$_B$ (*Liu et al., 1992*; *Vergnes et al., 1997*; *Bortolato et al., 2010*) receptors. Prior experimental and computational work has demonstrated that a shift from GABA$_A$ receptor-mediated to GABA$_B$ receptor-mediated inhibition at the RT-TC synapse transforms oscillations from a 10 Hz, sparse, spindle-like activity to a 3 Hz, hyper-synchronized, seizure-like state (*von Krosigk et al., 1993*; *Destexhe et al., 1996*; *Destexhe, 1998*; *Blumenfeld and McCormick, 2000*).

GABA transporters (GATs) powerfully control the activation of GABA$_B$ receptors (*Beenhakker and Huguenard, 2010*). GAT1 and GAT3 represent the primary GABA transporters expressed in the brain and normally recycle GABA from the extrasynaptic space, thereby regulating GABA spillover from the synapse and the activation of extrasynaptic GABA$_A$ and GABA$_B$ receptors (*Cope et al., 2009*; *Scanziani, 2000*). In the thalamus, the more abundant transporter, GAT3, is localized farther away from synapses than GAT1 (*De Biasi et al., 1998*; *Beenhakker and Huguenard, 2010*). Consequently, specific GAT1 blockade results only in an increase in the amplitude of the GABA$_B$ IPSC, reflecting increases in GABA concentration near the synapse. In contrast, specific GAT3 blockade results in an increase in both amplitude and decay of the GABA$_B$-mediated inhibitory post-synaptic current (GABA$_B$ IPSC), as GABA is allowed to diffuse far from the synapse where there is an abundance of GABA$_B$ receptors (*Kulik et al., 2002*; *Beenhakker and Huguenard, 2010*). On the other hand, dual GAT1 and GAT3 blockade results in a roughly 10-fold increase in the decay time constant of the GABA$_B$ IPSC, a supralinear effect relative to the modest increases for single GAT1 (no increase) or GAT3 (1.5-fold) blockade. These findings were replicated in a diffusion-based computational model (*Beenhakker and Huguenard, 2010*).

In this study, we investigate the consequences of physiologically relevant GABA$_B$ receptor-mediated inhibition observed during different combinations of GABA transporter blockade: control, GAT1 blockade, GAT3 blockade and dual GAT1+GAT3 blockade (*Beenhakker and Huguenard, 2010*). As absence seizures are dependent on GABA$_B$ receptor signaling, we hypothesized that GAT blockade would regulate both thalamocortical neuron rebound bursting and network-level thalamic oscillations. We examine the effects of different GABA$_B$ receptor activation waveforms on both absence seizure-like thalamic oscillations and single thalamocortical neuron responses. We first use pharmacological manipulations to demonstrate that individual GAT1 or GAT3 blockade prolongs seizure-like oscillations, but that dual GAT1+GAT3 blockade surprisingly abolishes oscillations. Next, we apply physiological GABA$_B$ IPSC waveforms corresponding to each pharmacological condition to single thalamocortical neurons with dynamic clamp and demonstrate that individual GAT1 or GAT3 blockade increases rebound burst probability, but that dual GAT1+GAT3 blockade suppresses it. We then build computational model neurons to explore how differential GABA$_B$ IPSC kinetics shape rebound bursting in TC neurons. These models show how prolonged inhibition that pushes instantaneous T channel inactivation states toward steady-state values is responsible for rebound burst failure, and thus the nonlinear effects of rebound bursting associated with dual GAT1+GAT3 blockade. Finally, we build computational thalamic network models to demonstrate that the same interplay between GABA$_B$-mediated inhibition and T channels is sufficient to explain the effects of GAT blockade on biological thalamic oscillations. Through these experimental and computational approaches, we identify how GABA$_B$-mediated inhibition across both voltage and time dimensions regulates T channel activity and seizure-like oscillations.

## Results

### Thalamic oscillations

To evaluate the contribution of GABA transporters to thalamic network activity in the context of $GABA_B$ receptor-mediated inhibition, we used a standard, acute rat brain slice model in which electrical oscillations are evoked by extracellular stimulation of the synaptic inputs to the reticular thalamic nucleus in the presence of the $GABA_A$ receptor blocker bicuculline (*Huguenard and Prince, 1994*; *Jacobsen et al., 2001*; *Kleiman-Weiner et al., 2009*). We evoked oscillations at intervals producing no rundown (once per minute, *Jacobsen et al., 2001*) and monitored neuronal activity with extracellular multiunit electrodes placed within the ventrobasal complex of the thalamus. By detecting evoked bursts, we found that oscillations last between 2 and 13 s at baseline (*Figure 1A*). The autocorrelogram of binned spike times revealed pronounced secondary peaks at multiples of approximately 500 ms (*Figure 1A*). After recording evoked oscillations for 20 min under baseline conditions, we then applied one of four experimental solutions to the perfusate (*Figure 1B*). Experimental solutions consisted of: (1) a control solution identical to the baseline solution, (2) 4 µM NO-711, a specific GAT1 blocker (*Sitte et al., 2002*), (3) 100 µM SNAP-5114, a specific GAT3 blocker (*Borden et al., 1994*), or (4) a combination of 4 µM NO-711 and 100 µM SNAP-5114. These blocker concentrations achieve full GAT blockade (*Beenhakker and Huguenard, 2010*). Experimental solutions were applied for 40 min, a duration sufficient for solution wash in and response stabilization (*Beenhakker and Huguenard, 2010*), and then washed out over another 20 min.

When individually applied, either GAT1 or GAT3 blockade prolonged oscillations (*Figure 1C*), consistent with the absence seizure exacerbation seen with a clinically used GAT1 blocker, tiagabine (*Ettinger et al., 1999*; *Knake et al., 1999*; *Vinton et al., 2005*). We measured the duration of each evoked oscillation, then computed the average duration of the last five stable oscillations in baseline and experimental solutions (*Figure 1D*). Relative to baseline, individual GAT1 or GAT3 blockade increased oscillation duration by 36% (n = 8 slices from five animals, p=0.0027; here and in all results, percentages refer to relative change from control conditions) and 99%, (n = 7 slices from five animals, p=0.0076), respectively. We also evaluated the effects of individual GAT1 or GAT3 blockade on the period of evoked oscillations (*Figure 1E*). Relative to baseline, blocking either GAT1 or GAT3 increased the oscillation period by 13% (n = 8 slices from five animals, p=0.0021) and 32% (n = 7 slices from five animals, p=0.0050), respectively. Collectively, the effects of GAT blockade on oscillation properties generally agreed with the previously reported actions of GAT blockers on isolated $GABA_B$ receptor-mediated IPSCs. That is, the 1.4-fold and 2-fold increase in oscillation duration corresponds roughly to the reported 1.5-fold and 2.2-fold increase in $GABA_B$ IPSC amplitude produced by GAT1 or GAT3 blockade, respectively (*Beenhakker and Huguenard, 2010*). Additionally, GAT3 blockade significantly prolonged oscillation period, while the effect for GAT1 blockade was modest, consistent with reported effects on isolated $GABA_B$ IPSC decay (*Beenhakker and Huguenard, 2010*).

Surprisingly, the effects of NO-711 and SNAP-5114 co-perfusion on evoked oscillations were not additive. Rather than prolonging evoked oscillations, dual GAT1+GAT3 blockade ultimately *eliminated* oscillations. Following a brief prolongation during the early phases of drug perfusion (see *Figure 1C*), dual GAT1+GAT3 blockade eventually decreased oscillation duration by 48% (*Figure 1D*, n = 9 slices from four animals, p=0.026). As the effects of dual blockade on oscillation duration did not appear to reach a steady state by 40 min, we extended the drug application to 60 min in a subset of experiments. For those slices, dual GAT1+GAT3 blockade invariably abolished oscillations (*Figure 1F*, n = 5 slices from three animals, p=0.0062).

In summary, the observed effects of individual GAT1 or GAT3 blockade on oscillation duration and period generally reflect the actions these individual blockers have on $GABA_B$ receptor-mediated IPSCs isolated from thalamocortical neurons. GAT blockade-dependent increases in IPSC amplitude were associated with increased strength of oscillation, as measured by duration. However, the effects of dual GAT1+GAT3 blockade on oscillation duration did not reflect the additive effects of combined blockade on $GABA_B$ IPSCs (*Beenhakker and Huguenard, 2010*). To better understand the discrepancy between GAT regulation of IPSCs and GAT regulation of thalamic oscillations, we next examined how IPSC amplitude and kinetics regulate the activity of single thalamocortical neurons.

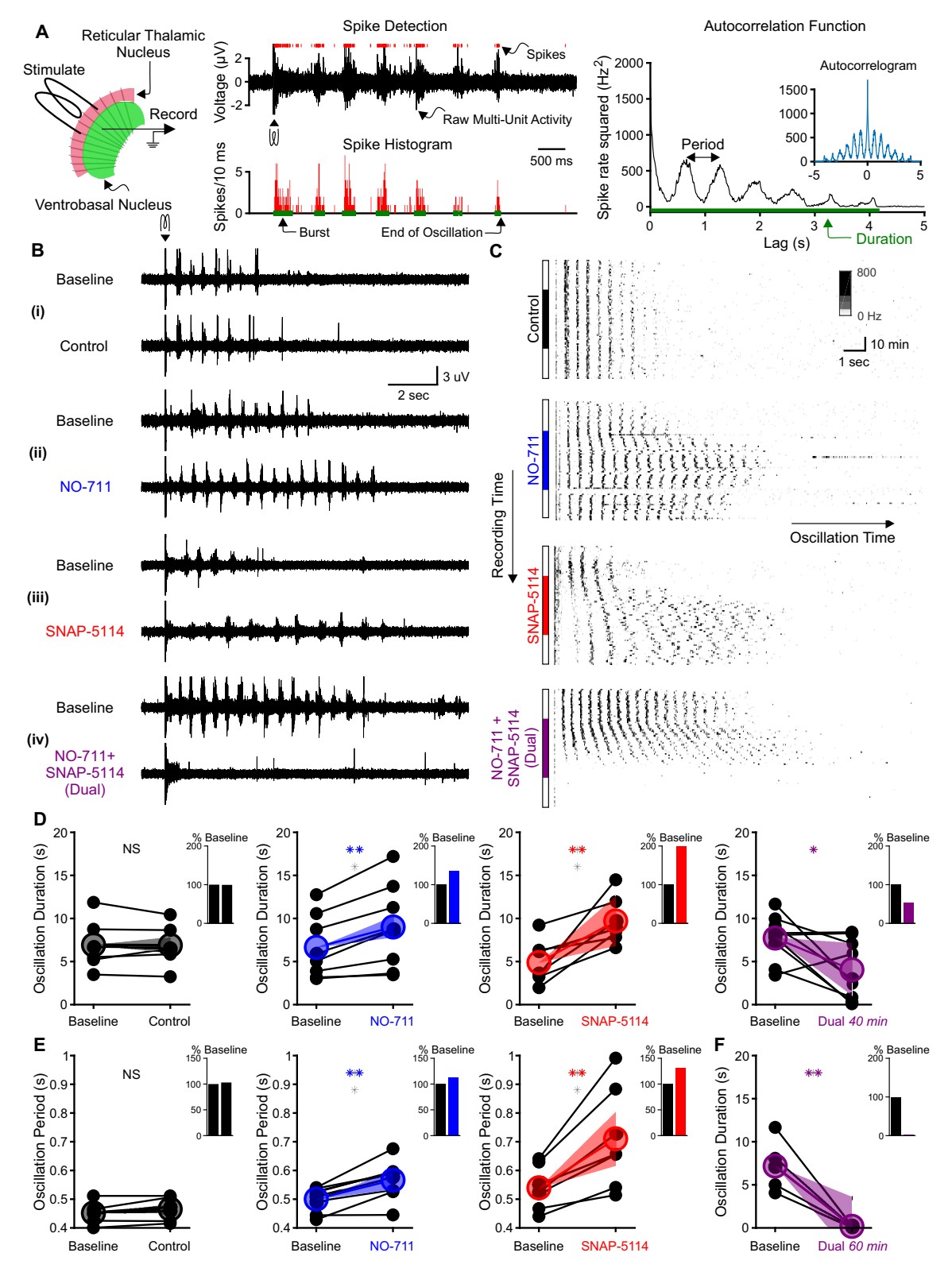

**Figure 1.** Individual GAT1 or GAT3 blockade strengthens thalamic oscillations, but dual GAT1+GAT3 blockade abolishes oscillations. (**A**) Slice recording setup and sample analysis. Acute thalamic slices were bathed in bicuculline to block GABA$_A$ receptors. A brief voltage stimulus (0.5 ms, 10 V) was applied with a bipolar electrode placed in either the reticular thalamic nucleus or the adjacent internal capsule to evoke epileptiform oscillations recorded extracellularly in the ventrobasal complex. Spikes were detected, binned, and grouped into bursts. The oscillation duration was measured

*Figure 1 continued on next page*

*Figure 1 continued*

from the spike histogram and the oscillation period was computed from the autocorrelation function of binned spikes. (**B**) Example evoked epileptiform oscillations at baseline and 40 min after perfusing with (i) control (no drug added), (ii) 4 µM NO-711 (GAT1 blocker), (iii) 100 µM SNAP-5114 (GAT3 blocker) or (iv) dual 4 µM NO-711+100 µM SNAP-5114. (**C**) Example PSTHs over the entire course of a single control, GAT1-, GAT3-, and dual-block experiment. Oscillations were evoked every 60 s, but only the first 17 s after stimulation are shown. Drugs were perfused for 40 min after 20 min of baseline, followed by 20 min of washout. (**D–F**) Oscillation measures for all slices. Colored circles denote the mean value, colored lines denote the mean change; and shaded areas denote the 95% confidence intervals for the mean change. (**D**) Oscillation duration did not change following control perfusion, but increased following NO-711 or SNAP-5114 perfusion. Dual NO-711+SNAP-5114 perfusion reduced oscillation duration (*$p < 0.05$, **$p < 0.01$, paired-sample *t*-test). Inset shows mean change relative to baseline. (**E**) Oscillation period did not change following control perfusion and lengthened following NO-711 or SNAP-5114 perfusion (**$p < 0.01$, paired-sample *t*-test). (**F**) After 60 min of dual NO-711 + SNAP-5114 perfusion, oscillations were abolished in all slices (**$p < 0.01$, paired-sample *t*-test).

The online version of this article includes the following source data for figure 1:

**Source data 1.** Oscillation measures in response to different GABA transporter blockade conditions.

## Single neuron recordings

We investigated the effects of GAT-modulated, GABA$_B$ receptor-mediated currents on thalamocortical neuron rebound bursting, as this property is likely critical for the initiation of each successive cycle of an evoked oscillation (*von Krosigk et al., 1993*; *Huguenard and Prince, 1994*; *Warren et al., 1994*). Experimentally evoked GABA$_B$ receptor-mediated IPSCs isolated in acute thalamic slices vary considerably in amplitude (*Beenhakker and Huguenard, 2010*), likely reflecting differences in synaptic activation of reticular thalamic neurons by the electrical stimulus. We therefore utilized an alternative approach to systematically examine the effects of GAT blockade on the firing properties of thalamocortical neurons: dynamic clamp (*Sharp et al., 1993*; *Ulrich and Huguenard, 1996*). We used GABA$_B$ receptor-mediated IPSC waveforms isolated under voltage clamp during each pharmacological condition (control, GAT1 blockade, GAT3 blockade, dual GAT1+GAT3 blockade) as conductance waveform commands applied to single thalamocortical neurons (*Figure 2A*). We refer to these dynamic clamp-mediated conductance waveforms as *dyn*IPSCs. We applied the *dyn*IPSC corresponding to each pharmacological condition (*dyn*Control, *dyn*GAT1-Block, *dyn*GAT3-Block, *dyn*Dual-Block; *Figure 2B*) to each recorded thalamocortical neuron.

Since neurons likely receive variable numbers of inhibitory inputs, we scaled the conductance amplitudes for each pharmacological *dyn*IPSC by 25%, 50%, 100%, 200%, 400% and 800%, yielding 24 possible *dyn*IPSC waveforms (i.e. four pharmacological conditions x six amplitude scales). Additionally, we delivered each of the 24 waveforms at three approximate holding potentials: −60 mV, −65 mV or −70 mV. Five non-consecutive repetitions were performed for each *dyn*IPSC waveform and holding potential condition. We found that differences across pharmacological *dyn*IPSCs were most significant when the amplitudes were scaled by 200%. Throughout our analyses, we compare across *dyn*IPSCs using this amplitude scale unless otherwise specified.

*Figure 2C* shows example responses to *dyn*IPSCs delivered with dynamic clamp. Post-inhibitory, low-threshold calcium spikes (LTS) frequently followed each *dyn*IPSC, with sodium-dependent action potentials often crowning each LTS. Herein, *LTS* refers to the slow, broad (~50 ms) event following inhibition. *Burst*, on the other hand, specifically refers to the collection of action potentials crowning the LTS. We quantified several properties of each post-inhibitory LTS and burst in response to each *dyn*IPSC, including the probability of occurrence and the latency from *dyn*IPSC onset (*Figure 2C*). We also computed LTS features such as peak voltage value, maximum rising slope and the number of spikes per LTS, averaged across trials for each neuron (*Figure 2—figure supplement 1A*).

We first examined LTS and burst probability distributions over all recorded neurons following delivery of *dyn*IPSCs (LTS: *Figure 2—figure supplement 1B–C*, burst: *Figure 2D–E*). Relative to *dyn*Control responses, LTS and burst probabilities were higher following either *dyn*GAT1-Block (n = 47 cells, LTS: +26%, p=0.0080, burst: +63%, p=0.0018) or *dyn*GAT3-Block (n = 47 cells, LTS: +39%, p=0.0015, burst: +106%, p=$1.7 \times 10^{-7}$), but lower following *dyn*Dual-Block (n = 47 cells, LTS: −88%, p=$4.8 \times 10^{-6}$, burst: −82%, p=0.030). We observed the same trend across pharmacological *dyn*IPSCs for all other conductance scales (LTS: *Figure 2—figure supplement 1C*, burst: *Figure 2E*). Not surprisingly, increasing the conductance scale increased LTS and burst probability following either *dyn*Control, *dyn*GAT1-Block or *dyn*GAT3-Block. However, both probabilities were consistently very low, below 6%, across all conductance scales following *dyn*Dual-Block. These changes in thalamocortical

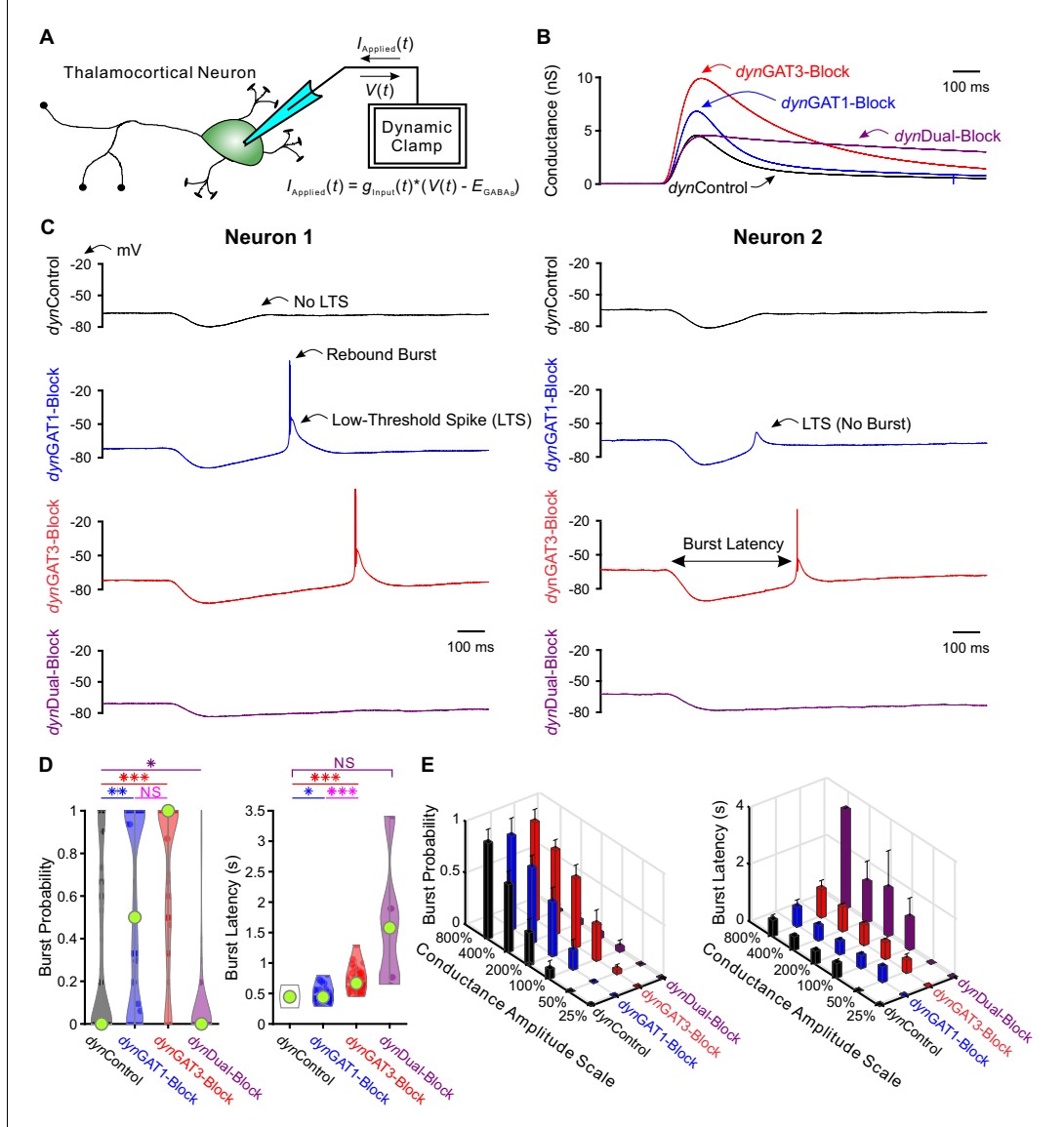

**Figure 2.** Post-inhibitory, low-threshold rebound spikes and bursts in thalamocortical neurons are bidirectionally modulated by GABA_B receptor-mediated conductance waveforms. (**A**) Dynamic clamp setup. A thalamocortical neuron was patched in the whole-cell configuration. The applied current was computed from the instantaneous voltage and a command conductance waveform over time to simulate GABA_B receptor activation. (**B**) Command GABA_B receptor conductance waveforms (*dyn*IPSCs, amplitudes scaled by 200%) that emulated different GAT blockade conditions based on GABA_B IPSCs isolated with voltage clamp (***Beenhakker and Huguenard, 2010***). (**C**) Sample voltage responses of two neurons to the four different *dyn*IPSCs shown in (**B**). (**D**) Distributions of post-inhibitory rebound burst measures over all 47 recorded neurons across *dyn*IPSCs shown in (**B**). For burst latency, only responsive neurons are included. Relative to *dyn*Control responses, rebound burst probability increased following either *dyn*GAT1- or *dyn*GAT3-Block, but decreased following *dyn*Dual-Block (*p<0.05, **p<0.01, ***p<0.001, Friedman's test for burst probability, repeated-measures ANOVA and paired-sample *t*-test for burst latency). (**E**) Mean burst measures over all 47 recorded neurons, across four different *dyn*IPSC waveforms and six different conductance amplitude scales. Error bars denote 95% confidence intervals.

The online version of this article includes the following source data and figure supplement(s) for figure 2:

**Source data 1.** LTS and burst features in response to IPSCs recorded with dynamic clamp (*dyn*IPSCs).

**Figure supplement 1.** Post-inhibitory, low-threshold rebound spikes and bursts in thalamocortical neurons are bidirectionally modulated by GABA_B receptor-mediated conductance waveforms.

neuron rebound burst probability parallel the prolonged oscillation duration following individual GAT1 or GAT3 blockade, and the decrease in oscillation duration following dual GAT1+GAT3 blockade (*Figure 1D*).

Next, we examined distributions of average LTS and burst latencies over neurons responsive to *dyn*IPSCs (LTS: *Figure 2—figure supplement 1B–C*, burst: *Figure 2D–E*). We restricted this analysis to *dyn*GAT1- and *dyn*GAT3-Block because *dyn*Dual-Block did not reliably evoke LTSs. Relative to *dyn*Control responses, average LTS latency was not significantly different following *dyn*GAT1-Block (n = 32 cells, p=0.97), while average burst latency was modestly prolonged (+4.6%, n = 21 cells, p=0.034). In contrast, *dyn*GAT3-Block IPSCs significantly prolonged both LTS (+53%, n = 32 cells, p=$3.7 \times 10^{-9}$) and burst latency (+58%, n = 21 cells, p=$1.1 \times 10^{-9}$). We observed the same trend across pharmacological conditions for all other conductance amplitude scales (*Figure 2E*). As the inter-burst interval (latency from last burst) separates each cycle of seizure-like oscillations, and is dominated by inhibition of TC cells (*Bal et al., 1995*), the above results are consistent with the increased oscillation period following either individual GAT1 or GAT3 blockade, but a more pronounced effect for the latter (*Figure 1E*). *Table 1* provides a summary of all LTS and burst features (see also *Figure 2—figure supplement 1B–C*).

In summary, the bidirectional differences in thalamocortical neuron rebound burst properties in response to different GABA$_B$ activation waveforms (i.e. burst enhancement with either GAT1 or GAT3 blockade, but burst elimination during dual blockade) were correlated with the bidirectional differences in thalamic oscillation duration and period during the corresponding pharmacological manipulations. That is, by ultimately regulating thalamocortical neuron bursting, GATs appear to powerfully control thalamic network oscillations through differential activation of GABA$_B$ receptors.

## Single neuron models

We next sought to determine the essential components of the thalamocortical neuron that contributes to the differential *dyn*IPSC responses observed during our dynamic clamp experiments. We also sought to better understand the underlying channel dynamics contributing to the differential responses. Toward these ends, we established a conductance-based, multi-compartment, single neuron model for each of the 36 experimentally recorded thalamocortical neurons for which we had stable responses across all acquired conductance amplitude scales (*Figure 3A*).

Our preliminary modeling results using existing TC cell models (*Destexhe et al., 1998*; *Amarillo et al., 2014*) failed to recapitulate two key features of GABA$_B$ receptor-mediated post-inhibitory rebound LTSs that are likely critical in determining network level responses. Notably, the average LTS latencies of the model responses were routinely much earlier (400 ~ 1000 ms) than the biological ones (400 ~ 4000 ms). In addition, the model LTS and burst responses were graded in amplitude as a function of inhibitory strength, in contrast to the more characteristic all-or-none responses of recorded neurons. Therefore, we developed a gradient descent fitting approach to obtain suitable multicompartment models compatible with the data. As prior computational and

**Table 1.** Average change of LTS and burst features relative to *dyn*Control responses.
For LTS and burst probability, Friedman's test with multiple comparison was used across all four groups. Due to the lack of LTS response to *dyn*Dual-Block, the comparison between *dyn*Control and *dyn*Dual-Block for all other LTS and burst features was conducted separately using the paired-sample *t*-test. Across *dyn*Control, *dyn*GAT1-Block and *dyn*GAT3-Block responses, Friedman's test with multiple comparison was used for LTS latency and repeated-measures ANOVA with multiple comparison was used otherwise.

|  | *dyn*GAT1-Block | *dyn*GAT3-Block | *dyn*Dual-Block |
|---|---|---|---|
| Burst Probability | +63%, p=0.0018 | +106%, p=$1.7 \times 10^{-7}$ | −82%, p=0.030 |
| Burst Latency | +4.6%, p=0.034 | +58%, p=$1.1 \times 10^{-9}$ | no change, p=0.066 |
| LTS Probability | +26%, p=0.0080 | +39%, p=0.0015 | −88%, p=$4.8 \times 10^{-6}$ |
| LTS Latency | no change, p=0.97 | +53%, p=$3.7 \times 10^{-9}$ | +254%, p=0.044 |
| Spikes Per LTS | +62%, p=$1.3 \times 10^{-6}$ | +93%, p=$9.3 \times 10^{-7}$ | no change, p=0.095 |
| LTS Peak Value (mV) | +2.5 ± 0.5 mV, p=$1.4 \times 10^{-4}$ | +3.3 ± 0.8 mV, p=$5.2 \times 10^{-4}$ | no change, p=0.33 |
| LTS Maximum Slope | +52%, p=$2.6 \times 10^{-9}$ | +82%, p=$1.4 \times 10^{-9}$ | no change, p=0.068 |

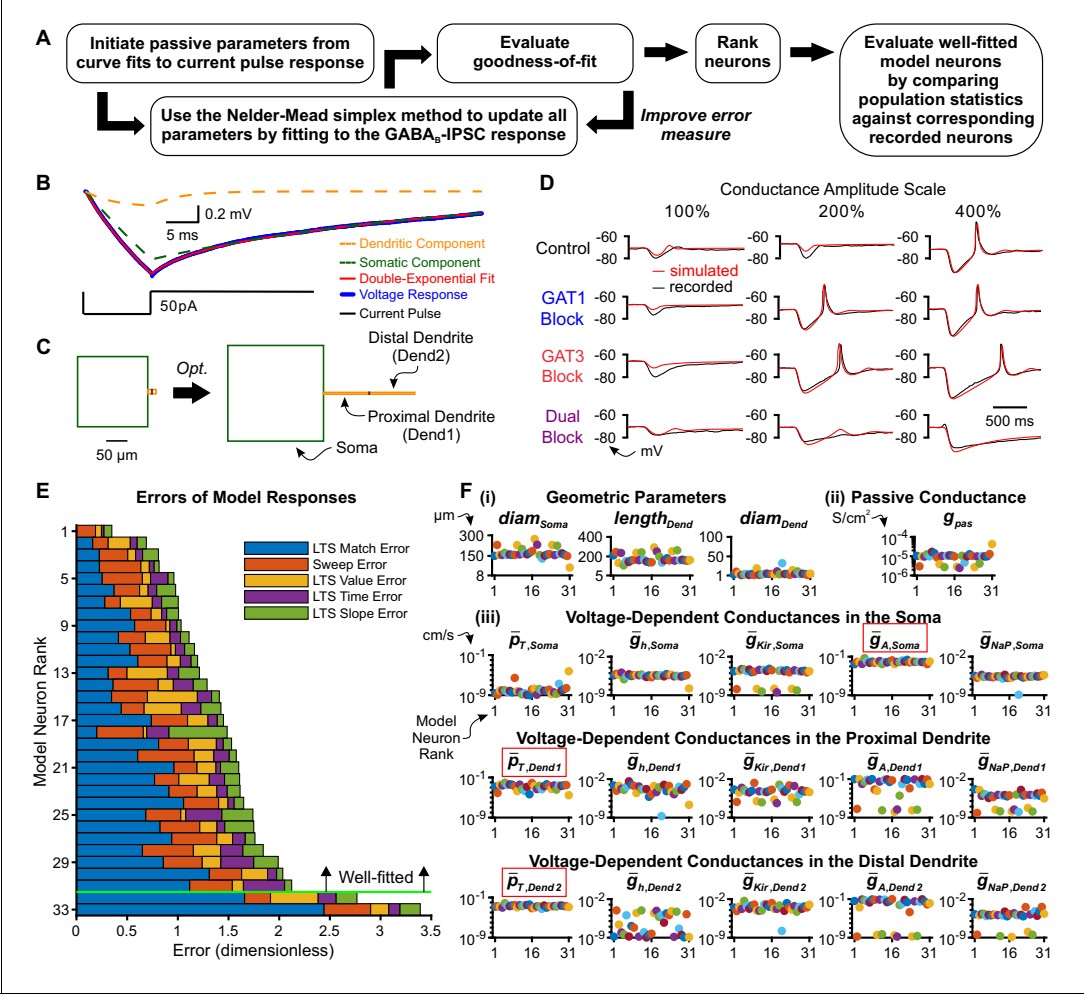

**Figure 3.** Model thalamocortical neurons reproduce GABA$_B$ IPSC and rebound responses. (**A**) Model optimization workflow. (**B**) Sample double-exponential curve fits (red) to averaged current pulse responses (blue) for an example neuron. The dashed lines correspond to the curves representing the somatic compartment (green) and dendritic compartment (orange). (**C**) Ball-and-stick geometries estimated from the two exponential components shown in (**B**) (*Johnston and Wu, 1994*) were converted to cylindrical, three compartment models (left), which were then optimized (right). (**D**) Fits of simulated *sim*IPSC responses to recorded *dyn*IPSC responses, for the same example neuron. (**E**) The 33 model neurons that underwent optimization were ranked by a weighted average of five different types of errors (see Materials and methods). The 31 highest ranked model neurons were considered *well-fitted*. The example neuron in (**D**) had rank 6. (**F**) Final values of parameters that could vary for the 31 well-fitted model neurons. Note that the T channel density is invariably high in the dendrites and the A-type potassium channel density is invariably high in the soma among all model neurons (red boxes). Geometric parameters are in μm, maximal conductance densities ($\bar{g}$) and conductance densities ($g$) are in S/cm$^2$ and maximal permeability densities ($\bar{p}$) are in cm/s. The x axis is the model neuron rank in (**E**). We distinguish among (**i**) geometric parameters, (**ii**) voltage-independent *passive* parameters and (**iii**) voltage-dependent *active* parameters.

The online version of this article includes the following source data and figure supplement(s) for figure 3:

**Source data 1.** Parameters and errors for optimized model thalamocortical neurons.

**Figure supplement 1.** Additional fits of simulated *sim*IPSC responses to recorded *dyn*IPSC responses for selected neurons.

experimental work demonstrates the importance of higher T channel densities in dendritic versus somatic compartments (*Destexhe et al., 1998*; *Munsch et al., 1997*; *Williams and Stuart, 2000*; *Zhou et al., 1997*), we modeled each thalamocortical neuron by a cylindrical somatic compartment and two cylindrical dendritic compartments in series (*Figure 3C*).

To reduce the number of fitted parameters, some simplifying assumptions were made: the somatic length and diameter were equivalent, the two dendritic compartments had equal dimensions, and passive leak channels were inserted into all three compartments at uniform densities. Four voltage-independent (passive) parameters were allowed to vary across model neurons: the somatic

diameter ($diam_{soma}$), the dendritic diameter ($diam_{dend}$), the dendritic length ($length_{dend}$) and the passive leak conductance density ($g_{pas}$). As prior work has identified ionic currents that contribute to the resting membrane potential of thalamocortical neurons (*Amarillo et al., 2014*), we inserted the following five voltage-dependent channels in all three compartments: the T-type calcium channel ($I_T$), the hyperpolarization-activated nonspecific cationic channel ($I_h$), the A-type transient potassium channel ($I_A$), the inward-rectifying potassium channel ($I_{Kir}$) and the persistent sodium channel ($I_{NaP}$). The densities of voltage-dependent channels were allowed to vary across compartments, resulting in a total of 15 (5 currents x 3 compartments) voltage-dependent (active) parameters that were allowed to vary across model neurons.

To provide an initial estimate of the geometric parameters that corresponded to each recorded thalamocortical neuron, we applied the short pulse methodology described by Johnston and Wu (1994, Chapter 4). During dynamic clamp experiments, we applied a short current pulse at the beginning of each recorded sweep. The average current pulse response for each neuron was well-fitted by a double exponential function (*Figure 3B*). From the coefficients and time constants of the two exponential components, we inferred the following four parameters for a ball-and-stick model (*Rall, 1962*): input conductance, electrotonic length, dendritic-to-somatic conductance ratio and the membrane time constant. These Rall model values were then converted into initial passive parameter seed values ($diam_{Soma}$, $diam_{Dend}$, $length_{dend}$, $g_{pas}$) of each three-compartment model neuron (see Materials and methods). Notably, the average electrotonic length across all recorded neurons was 0.71 ± 0.08 space constants (range: 0.1-1.5 space constants), mitigating voltage attenuation concerns that might arise from applying currents through the soma with dynamic clamp.

Single thalamocortical neuron responses recorded during dynamic clamp experiments served to optimize passive and active parameters of each three-compartment, model thalamocortical neuron. GABA$_B$ receptors were placed in the somatic compartment of each model neuron, and activation waveforms identical to the conductance waveforms used in dynamic clamp (i.e. *dyn*IPSCs) were applied. We refer to these simulated GABA$_B$ receptor activation waveforms as *sim*IPSCs, corresponding to each pharmacological condition (*sim*Control, *sim*GAT1-Block, *sim*GAT3-Block, *sim*Dual-Block). For each model neuron, responses to *sim*IPSCs were iteratively compared to recorded *dyn*IPSC responses. We evaluated the goodness-of-fit for each iteration with a total error defined by a weighted combination of component errors (see Materials and methods). Examples of resultant geometry and voltage response fits are shown in *Figure 3C and D*, respectively.

Each model neuron was trained using a set of 12 recorded traces, each selected from a different *dyn*IPSC waveform, but evaluated against all 60–180 recorded traces for the neuron and ranked by the total error (*Figure 3E*). By removing neurons with a total error greater than two standard deviations above the mean, we designated the top 31 neurons as the set of *well-fitted model neurons*. Voltage response fits for selected well-fitted and excluded neurons are shown in *Figure 3—figure supplement 1*. All well-fitted neurons were characterized by high T channel densities in the distal dendrite and high A-type potassium channel densities in the soma (*Figure 3F*). Considerable variability among model neurons was observed in the densities of other ionic channels, likely contributing to the heterogeneity in LTS and burst measures among recorded neurons in response to each GABA$_B$ *dyn*IPSC waveform (*Figure 2D*).

We compared the distribution of LTS probabilities and features over the 31 well-fitted model neurons to the corresponding distributions derived from their biological counterparts recorded in dynamic clamp. In general, there was high agreement between the model simulations and dynamic clamp recordings (*Figure 4A and B*). Relative to *sim/dyn*Control responses, LTS probability was increased following *sim/dyn*GAT1-Block (n = 31, simulation: +30%, p=0.0086, dynamic clamp: +31%, p=0.038) or *sim/dyn*GAT3-Block (simulation: +44%, p=7.4×10$^{-4}$, dynamic clamp: +45%, p=0.011) and decreased following *sim/dyn*Dual-Block (simulation: −75%, p=9.8×10$^{-7}$, dynamic clamp: −93%, p=5.7×10$^{-4}$). Relative to *sim/dyn*Control responses, average LTS latency was not different following *sim/dyn*GAT1-Block (simulation: n = 27, p=0.85, dynamic clamp: n = 21, p=0.99) but was increased following *sim/dyn*GAT3-Block (simulation: +42%, n = 27, p=7.3×10$^{-8}$, dynamic clamp: +52%, n = 21, p=2.4×10$^{-6}$). Differences in average number of spikes per LTS, average LTS peak value and average LTS maximum slope across *dyn*IPSC waveforms were not sufficiently captured by model neurons (*Figure 4—figure supplement 1*). The same trends across pharmacological conditions were observed for all other conductance amplitude scales, showing high agreement

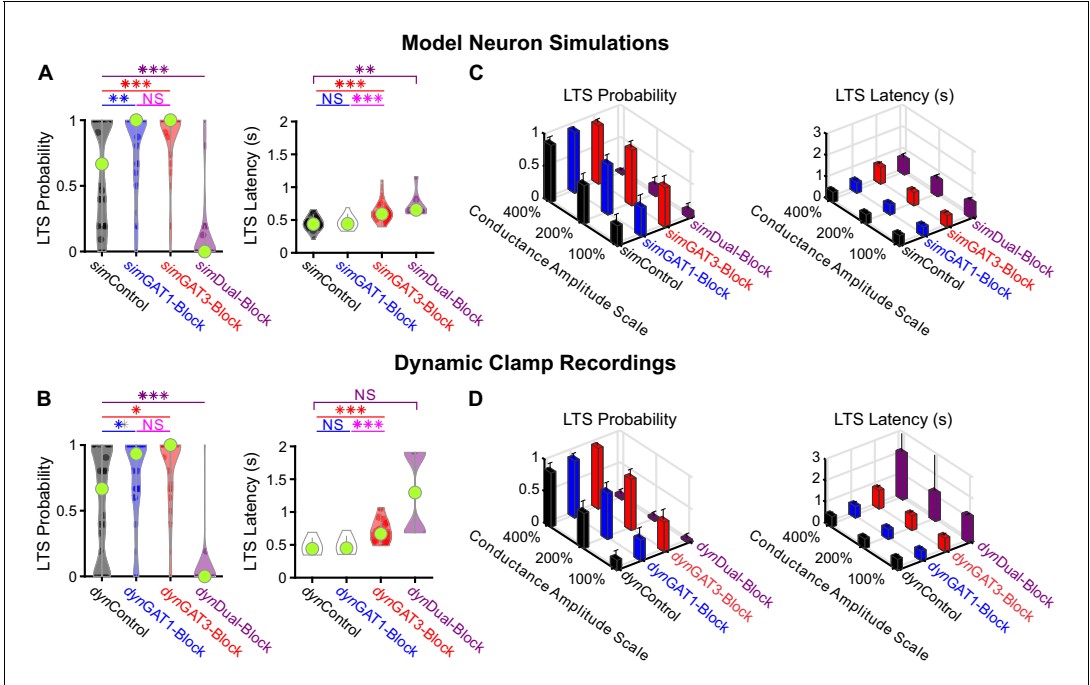

**Figure 4.** Well-fitted model and recorded neurons show similar low-threshold rebound spike probabilities and latencies in response to different GABA<sub>B</sub> IPSC waveforms. (**A**) Distributions of post-inhibitory, low-threshold rebound spike measures over the 31 well-fitted model neurons across GABA<sub>B</sub> IPSC waveforms shown in *Figure 2B* (*p<0.05, **p<0.01, ***p<0.001, repeated-measures ANOVA for LTS probability, Friedman's test for LTS latency). (**B**) Same as (**A**) but for the corresponding 31 recorded neurons (Friedman's test). (**C**) Mean low-threshold rebound spike measures over all 31 model neurons, across four different GABA<sub>B</sub> IPSC waveforms and three different conductance amplitude scales. Error bars denote 95% confidence intervals. (**D**) Same as (**C**) but for the corresponding 31 recorded neurons.

The online version of this article includes the following source data and figure supplement(s) for figure 4:

**Source data 1.** Simulated and recorded LTS and burst features across well-fitted neurons.

**Figure supplement 1.** Comparison of other low-threshold rebound spike or rebound burst measures between well-fitted model and recorded neurons.

between model and recorded neurons for LTS probability and latency, but not for other LTS features (*Figure 4C and D*, *Figure 4—figure supplement 1*).

In summary, many single neuron models were established that sufficiently recapitulated the probability and timing of post-inhibitory rebound bursts in response to 12 different physiological GABA<sub>B</sub>-receptor IPSC waveforms. A commonality among well-fitted model neurons is that T channel densities were high in the dendrites and A-type potassium channel densities were high in the soma, while there was heterogeneity in other channel densities.

## Interplay between GABA<sub>B</sub> receptors and T-type calcium channels

We next sought to understand the underlying mechanisms contributing to the different burst responses following different GAT-modulated, GABA<sub>B</sub>-mediated IPSCs. The IPSC-evoked, post-inhibitory rebound LTS has been shown to be caused by T-type calcium currents (*Llinás and Jahnsen, 1982*; *Kim et al., 2001*, *Figure 5—figure supplement 1A–C*). While much is known regarding the gating properties of T channels, little is known how these properties behave in response to physiological inhibition. Although it has been proposed from artificial voltage ramp studies that rebound bursting is sensitive to the slope of voltage depolarization (*Gutierrez et al., 2001*), the underlying T channel dynamics that confer such voltage sensitivity remain unknown.

### Low-threshold spikes are produced when T channel open probability discrepancy passes a threshold

To understand how IPSCs shape T channel dynamics, we first compared examples of LTS-*producing* responses evoked by *sim*GAT1- and *sim*GAT3-Block waveforms with examples of LTS-*lacking*

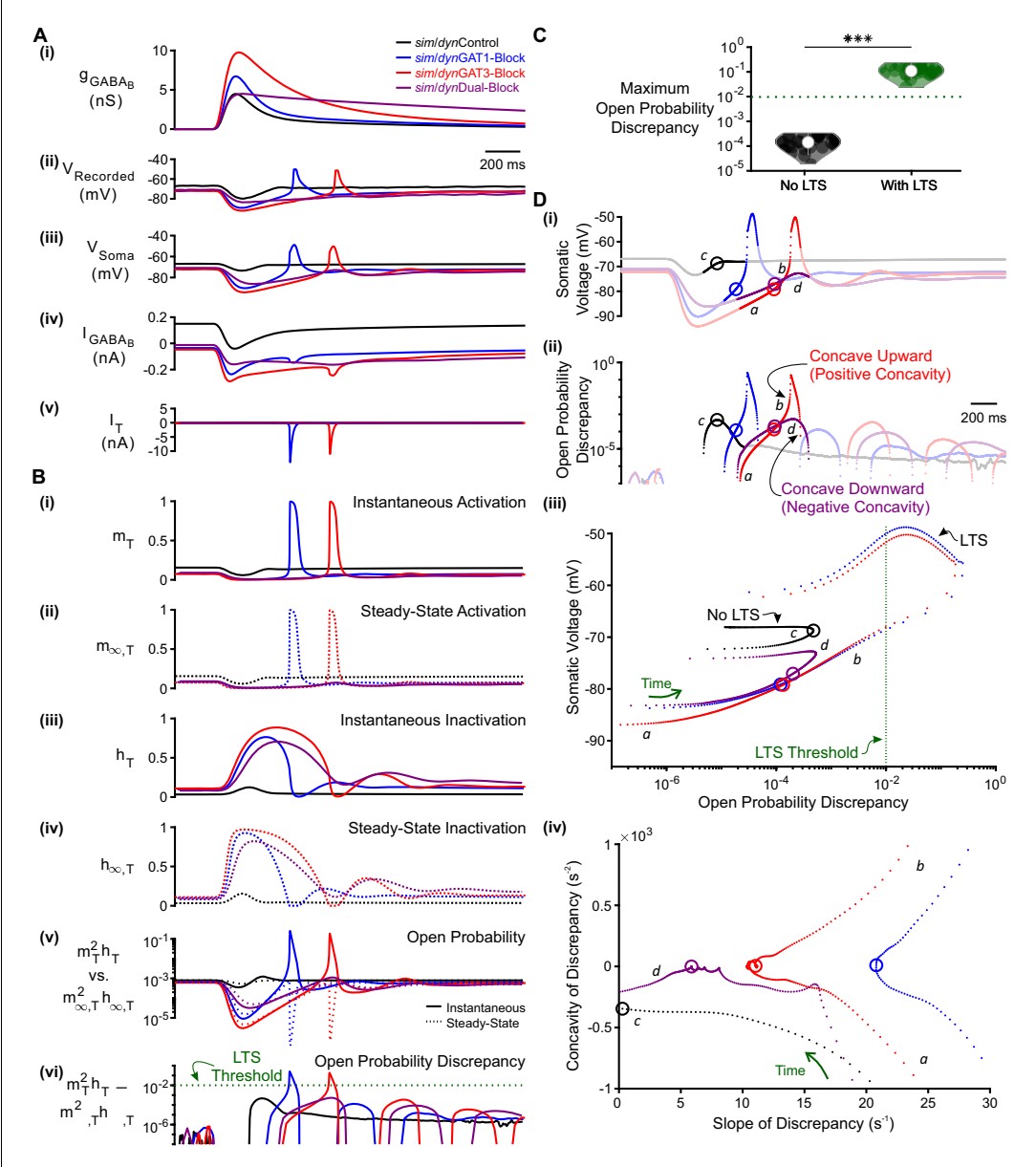

**Figure 5.** T-type calcium channel open probability discrepancy predicts LTS production following GABA_B IPSC waveforms. (**A**) The LTS response was correlated with the presence of large T-type calcium currents. (**i**) Command *sim/dyn*IPSCs as in **Figure 2B**. (**ii**) Voltage responses of Neuron 1 of **Figure 2C** recorded using dynamic clamp. (**iii-v**) Simulated responses of the corresponding model neuron (same as **Figure 3D**), including: (**iii**) somatic voltage, (**iv**) GABA_B receptor current, (**v**) T current. Currents were summed over all three compartments. (**B**) A combination of T channel activation (high $m_T$), T channel recovery (high $h_T$) and inactivation lag ($h_T$ different from $h_{T,\infty}$) appeared to be necessary for T channel opening ($m_T^2 h_T$ much higher than $m_{T,\infty}^2 h_{T,\infty}$). State variables for the distal dendritic T channel (other two compartments are similar), including: (**i**) instantaneous activation gating variable, (**ii**) steady-state activation gating variable, (**iii**) instantaneous inactivation gating variable, (**iv**) steady-state inactivation gating variable, (**v**) instantaneous open probability (solid line) versus steady-state open probability (dotted line), (**vi**) difference of instantaneous versus steady-state open probability (open probability discrepancy). Note that only positive discrepancies are plotted. The green dotted line here [also in (**C**) and (**D**) (**iii**) ] denotes the observed LTS threshold. (**C**) The maximum open probability discrepancy was higher in LTS-producing responses (***p<0.001, n = 31 cells, paired-sample *t*-test). (**D**) Within the first voltage peak following hyperpolarization highlighted in both (**i**) somatic voltage curves and (**ii**) dendritic T channel open probability discrepancy curves, LTS-producing responses followed trajectories different from LTS-lacking responses in either (**iii**) the voltage vs. open probability discrepancy phase plot or (**iv**) the concavity versus slope of discrepancy phase plot. The slope and concavity are the first and second derivatives of log_10(open probability discrepancy), respectively. Sample points plotted are 1 ms apart. Circles denote the decision points where either zero (or maximal negative) concavity is reached in the open probability discrepancy curves. Labels a, b, c, d are explained in the text.

The online version of this article includes the following source data and figure supplement(s) for figure 5:

*Figure 5 continued on next page*

*Figure 5 continued*

**Source data 1.** Open probability discrepancy measures for all simulated traces.
**Figure supplement 1.** Detailed analysis of factors contributing to LTS production following GABA$_B$ IPSCs.

responses evoked by *sim*Control and *sim*Dual-Block waveforms in a well-fitted model neuron (*Figure 5A*). We tracked the activation ($m_T$) and inactivation ($h_T$) gating variables of the T channel, as a function of time (*Figure 5B*). By convention, $m_T = 1$ when all channels are activated, and $h_T = 0$ when all channels are inactivated (*Hodgkin and Huxley, 1952*). Also by convention, T channel open probability is given by $m_T^2 h_T$ (*Huguenard and McCormick, 1992*). We also distinguished between steady-state values $[m_{T,\infty}(V)$ and $h_{T,\infty}(V)]$ that depend only on voltage, from instantaneous values $[m_T(V,t)$ and $h_T(V,t)]$ that depend on both voltage and time. At every time point, instantaneous values attempt to reach steady-state values exponentially through voltage-dependent time constants $[\tau_m(V)$ and $\tau_h(V)]$.

One notable feature of the T channel is that the activation time constant $\tau_m$ is rapid, indicating that the instantaneous activation variable $m_T$ quickly achieves its voltage-dependent steady-state value $m_{T,\infty}$. Thus, throughout the course of the relatively slow membrane voltage changes associated with the GABA$_B$-receptor-mediated IPSC, $m_T$ was nearly identical to $m_{T,\infty}$ [*Figure 5B(i–ii)*]. By contrast, the T channel inactivation time constant $\tau_h$ is slow, with a value about 10-fold higher than $\tau_m$ (*Coulter et al., 1989*). Thus, during the IPSC, the instantaneous T channel inactivation variable $h_T$ rarely achieved its voltage-dependent steady-state value $h_{T,\infty}$ [*Figure 5B(iii) and (iv)*].

At each time point, the capacity of $h_T$ to achieve $h_{T,\infty}$ can be quantified by calculating the difference between the two values. If the discrepancy between the two values is low (i.e. $h_T - h_{T,\infty} \approx 0$), then $h_T$ has successfully achieved its steady-state value at that time point. A high discrepancy, on the other hand, reflects an $h_T$ that has incompletely reached steady state. Importantly, the magnitude of discrepancy between $h_T$ and $h_{T,\infty}$ affected the T channel open probability $m_T^2 h_T$, but only when the activation variable $m_T$ was high. We therefore quantified the difference between instantaneous versus steady-state open probability ($m_T^2 h_T - m_{T,\infty}^2 h_{T,\infty}$), a parameter we simply refer to as *T channel open probability discrepancy* [*Figure 5B(vi)*]. When all 31 well-fitted model neurons were considered, the maximum T channel open probability discrepancy $\left[\max_t\left(m_T^2 h_T - m_{T,\infty}^2 h_{T,\infty}\right)\right]$ was on average 2.9 orders of magnitude higher for LTS-producing responses than for LTS-lacking responses (*Figure 5C*, n = 31 cells, p = 2.4 x $10^{-29}$). In fact, when all traces were considered, a threshold open probability discrepancy of $10^{-2}$ separated LTS-producing responses from LTS-lacking responses (not shown, herein referred to as the observed *LTS threshold*).

## Two criteria drive open probability discrepancy past LTS threshold

As high T channel open probability discrepancy predicted an LTS response, we sought to determine criteria that allowed this measure to pass LTS threshold (*Figure 5D*). For all *sim*IPSC conditions, when voltage depolarized during the decay phase of the IPSC, open probability discrepancy increased as $h_T$ (which was high from the hyperpolarization) slowly approached low $h_{T,\infty}$ (*Figure 5D (ii),(iii)*: label *a*); discrepancy was largely defined by the slow kinetics of $h_T$ because $m_T$ matches $m_{T,\infty}$ at all time points. However, near the end of the IPSC, the open probability discrepancy curve for LTS-producing responses reached an inflection point (point of zero concavity, *Figure 5D(ii),(iv)*: circles) that progressed towards positive concavity (concave upward, *Figure 5D(ii),(iv)*: label *b*) to eventually pass LTS threshold [blue and red curves]. In contrast, open probability discrepancy curves in LTS-lacking responses either failed to reach an inflection point [*Figure 5D(ii),(iv)*: black curve, label *c*], or reached an inflection point but veered back towards negative concavity [concave downward, *Figure 5D(ii),(iv)*: purple curve, label *d*]. In either case, the response failed to reach LTS threshold. We define the time point at which the open probability discrepancy curve reaches zero (or maximal negative) concavity as the *decision point* (circles in *Figure 5D*).

A comparison between the *sim*GAT3-Block (i.e. LTS-producing, red) and *sim*Dual-Block (i.e. LTS-lacking, purple) responses showed that at the decision point, the *slope* of open probability discrepancy, rather than its value, predicted whether positive or negative concavity was ultimately achieved [*Figure 5D(iv)*, circles]. A rapidly changing open probability discrepancy (i.e., steep slope) causes a

greater acceleration (i.e. positive concavity) in this measure. In fact, for all open probability curves reaching zero concavity, the slope of the open probability discrepancy curve at the decision point was always higher for LTS-producing responses than for LTS-lacking responses. We show the progression of trajectories aligned to the decision points for two conditions (*Video 1*, *sim*GAT3-Block and *sim*Dual-Block) and for all conditions (*Video 2*). When all 31 well-fitted model neurons were considered, there was a significant difference between LTS-producing and LTS-lacking responses for either the maximum open probability discrepancy concavity (*Figure 5—figure supplement 1D*), the

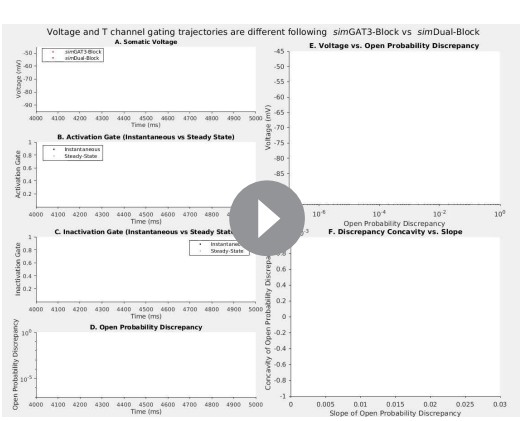

**Video 1.** Voltage and T-type calcium channel open probability discrepancy trajectories in response to *sim*GAT3-Block and *sim*Dual-Block GABA_B IPSCs. Simulated traces for the example neuron of *Figure 3*, resampled at 1 ms intervals, in response to either *sim*GAT3-Block (red) or *sim*Dual-Block (purple), as shown in *Figure 5*. Traces over simulation time are shown for (A) somatic voltage, (B) distal dendritic T channel activation gating variable, (C) distal dendritic T channel inactivation gating variable and (D) distal dendritic T channel open probability discrepancy. (E) Phase plot of somatic voltage versus distal dendritic T channel open probability discrepancy. (F) Phase plot of the concavity versus slope of the distal dendritic T channel open probability discrepancy. Note that only positive discrepancy values are plotted. All traces are aligned in Video time to the decision points (circles), defined as the point at which the open probability discrepancy curve either reaching an inflection point (zero concavity) or maximal negative concavity. Note that the traces for instantaneous and steady-state activation gates in (B) overlap on this time scale. In (E-F), only the first voltage peak following hyperpolarization is shown. The time points shown correspond to those with larger marker sizes in (A-D). Following either *sim*GAT3-Block or *sim*Dual-Block, the open probability discrepancy curve reaches an inflection point with zero concavity. However, only following *sim*GAT3-Block does the open probability discrepancy curve reach a significantly positive concavity, drive past the LTS threshold ($10^{-2}$) and produce a rebound low-threshold spike (LTS). Note that the slope of the open probability discrepancy curve at the decision point is higher for *sim*GAT3-Block than for *sim*Dual-Block.
https://elifesciences.org/articles/59548#video1

discrepancy slope at the decision point (*Figure 5—figure supplement 1E*) or the voltage slope at the decision point (*Figure 5—figure supplement 1F*). We conclude that an LTS is produced only if two criteria are satisfied: (1) the open probability discrepancy curve reaches an inflection point (e.g. *Figure 5D(iv)*: blue, red and purple curves), and (2) the slope of the open probability discrepancy curve at that inflection point is high (e.g. *Figure 5D(iv)*: blue and red curves).

## Open probability discrepancy depends on T channel inactivation kinetics

To test the contribution of T channel inactivation kinetics to LTS production, we bidirectionally altered the T channel inactivation time constant ($\tau_h$) for the same *sim*IPSC response simulations

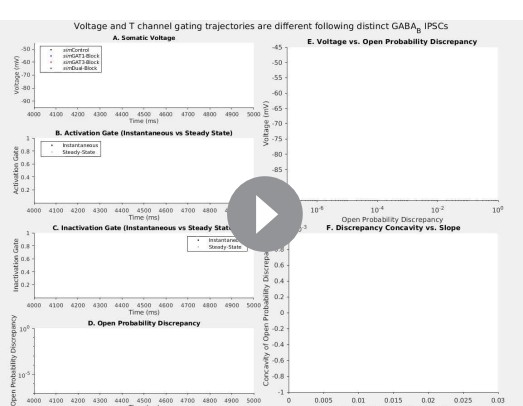

**Video 2.** Voltage and T-type calcium channel open probability discrepancy trajectories in response to all pharmacological GABA_B IPSCs. Same as *Video 1* but in response to all *sim*IPSCs shown in *Figure 5*. (A-F) See descriptions for *Video 1*. Note that following *sim*Control (black), the maximum concavity for the open probability discrepancy curve is negative, so its trajectory in the open probability discrepancy concavity versus slope phase plot (F) never reaches above the y = 0 line. For other *sim*IPSCs, note that the slope of the open probability discrepancy curve at the decision point is higher for LTS-producing trajectories [*sim*GAT1-Block (blue) and *sim*GAT3-Block (red)] than for an LTS-lacking trajectory [*sim*Dual-Block (purple)].
https://elifesciences.org/articles/59548#video2

as in *Figure 5*. When $\tau_h$ was halved in *sim*GAT1- and *sim*GAT3-Block simulations, T channel open probability discrepancy remained less than LTS threshold and LTS responses normally observed during GAT1 and GAT3 blockade were abolished (*Figure 6A* with *Figure 5A–B*). In contrast, doubling $\tau_h$ in *sim*Dual-Block simulations drove T channel open probability discrepancies past LTS threshold and, consequently, normally absent LTSs appeared (*Figure 6B* with *Figure 5A–B*). Note that *sim*-Control waveforms continued to result in LTS failure as voltage hyperpolarization was weak and T channel recovery was low.

## Open probability discrepancy depends on IPSC kinetics

To test the contribution of inhibition kinetics to T channel open probability discrepancy, we systematically varied the time constant of IPSCs while fixing the amplitude. Prolonging an LTS-producing *sim*GAT3-Block IPSC decreased T channel open probability discrepancy and abolished LTS responses as time constants increased above four-fold (*Figure 7A1*). Conversely, shortening a non-LTS-producing *sim*Dual-Block IPSC increased T channel open probability discrepancy and produced LTS responses as time constants decreased by 20% (*Figure 7A2*). We show the progression of trajectories aligned to the decision points for two time constants (*Video 3*, last LTS success and first LTS failure) and for all time constants (*Video 4*). As changing the kinetics of inhibition also changes the total amount of inhibition delivered to a cell, we also changed inhibition kinetics while keeping charge (i.e. the area under the curve) constant. Nonetheless, we continued to observe a decrease in T channel open probability discrepancy and eventual LTS failure as *sim*IPSC kinetics slowed (*Figure 7—figure supplement 1*). Therefore, inhibition kinetics appear to be important for LTS production through its influence on T channel open probability discrepancy.

Since hyperpolarization promotes T channel recovery (*Coulter et al., 1989*), it remains possible that sufficiently strong hyperpolarization – regardless of waveform – will produce an LTS. To test this possibility, we varied the inhibition amplitude using either the *sim*GAT3-Block (fast kinetics) or *sim*Dual-Block (slow kinetics) waveform. As we increased the amplitude of the *sim*GAT3-Block waveform while fixing the rise and decay time constants, LTSs emerged as T channel open probability discrepancy increased (*Figure 7B1*). In contrast, as the amplitude of the *sim*Dual-Block waveform increased while fixing the rise and decay time constants, T channel open probability discrepancy nonetheless remained low and robust LTS responses never emerged (*Figure 7B2*). Thus, fast inhibition kinetics are important for driving T channel open probability discrepancy, and slow kinetics provides an explanation for the consistently low LTS and burst probability across all conductance amplitude scales following *dyn*/*sim*Dual Block (*Figures 2E*, *4C and D*).

In summary, LTS production following physiological inhibition is largely controlled by the dynamics of T channel open probability discrepancy. We show that LTS production appears to depend not only on the *amplitude* of inhibition, but also the *temporal envelope* of inhibition. Large inhibition amplitude is required for sufficient T channel recovery, whereas fast inhibition decay is required for driving the T channel open probability discrepancy beyond an inflection point and creating a brief time window with sufficiently high T channel open probability for LTS production.

## Network models

We next explored whether the interplay between GABA$_B$-mediated inhibition and T type calcium channel dynamics in thalamocortical neurons contributes to the observed changes in network-level oscillations following GAT blockade. We first examined the effects of GABA$_B$ receptor-mediated inhibition in a simplified two-cell network configuration. In each two-cell network, we connected a single compartment, GABAergic reticular thalamic model neuron (*Klein et al., 2018*) to one of the 31 well-fitted model thalamocortical neurons (*Figure 8A*). To generate action potentials, Hodgkin-Huxley type sodium and potassium channels were inserted into the somatic compartment of each model neuron (*Williams and Stuart, 2000*). The reticular thalamic neuron was connected to the thalamocortical neuron only via a GABA$_B$ receptor-mediated inhibitory synapse (GABA$_A$ receptors were blocked during experimentally evoked oscillations, see *Figure 1*). Consistent with previous intra-thalamic models (*Destexhe et al., 1996*), the thalamocortical neuron provided AMPA receptor-mediated excitation to reticular thalamic neurons. By applying a brief stimulating current to the reticular thalamic neuron and varying the GABA$_B$ receptor activation parameters (*sim*IPSCs), GABA$_B$ conductance waveforms comparable to those used by dynamic clamp were evoked in each thalamocortical

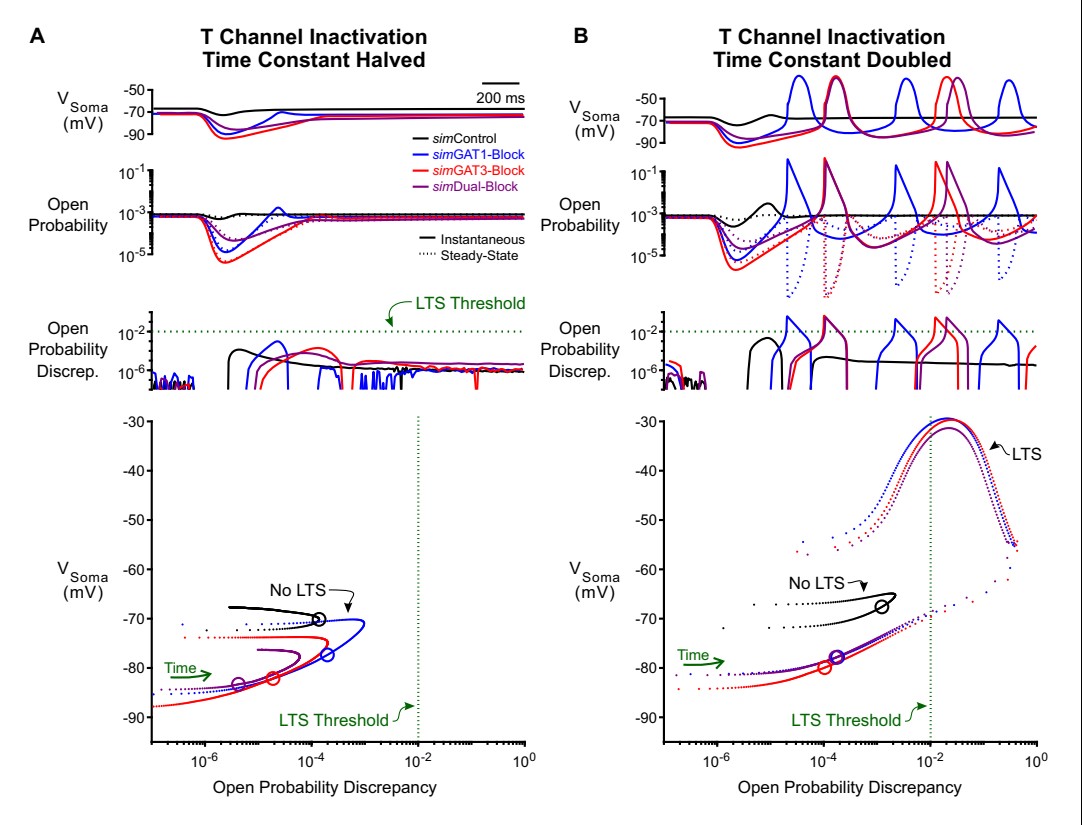

**Figure 6.** Manipulation of the T-type calcium channel inactivation time constant modulates LTS production. (**A**) Simulated responses as in *Figure 5* with the T channel inactivation time constant $\tau_{h_T}$ halved. The T channel open probability discrepancy failed to reach threshold and no LTS was produced following any *sim*IPSC waveform. The green dotted line denotes the observed LTS threshold from *Figure 5C*. In the voltage vs. open probability discrepancy phase plot, only the first voltage peak following hyperpolarization is shown. (**B**) Simulated responses as in *Figure 5* with $\tau_{h_T}$ doubled. The T channel open probability discrepancy increased and LTSs appeared following *sim*Dual-Block. Note that there is still no LTS produced following *sim*Control.

neuron (*Figure 8B*). To simulate variable resting membrane potentials of thalamocortical neurons, we varied the thalamocortical neuron leak reversal potential between −73 and −60 mV (14 leak reversal potentials). To generate trial-to-trial variability (five trials per leak reversal potential), we randomized the leak conductance of each neuron to within 10% of the original value. Of all 31 possible two-cell networks, 24 had a quiescent, pre-stimulation baseline over this range of leak reversal potentials. In those networks with quiescent baseline, oscillations persisted only when the GABA$_B$-mediated inhibition promoted T channel open probability discrepancy in thalamocortical neurons to produce rebound bursting (*Figure 8C*), consistent with our single neuron models. An oscillation probability was computed for each two-cell network over the 14 × 5 = 70 trials. In addition, an oscillatory period and an oscillatory index based on the autocorrelation function of pooled spikes was computed for each successfully evoked oscillation (see Materials and methods).

We examined the distributions of oscillation probability, average oscillation period and average oscillatory index over the 24 different two-cell networks (*Figure 8D*). Relative to using *sim*Control parameters, oscillation probability increased when using *sim*GAT1-Block parameters (+66%, n = 24 networks, p=0.016) or *sim*GAT3-Block parameters (+93%, p=0.048). These results are consistent with the experimental observation that individual GAT1 or GAT3 blockade prolonged oscillations (*Figure 1D*). Relative to using *sim*Control parameters, average oscillation period increased when using *sim*GAT3-Block parameters (+39%, n = 10, p=4.2×10$^{-4}$), also consistent with the experimental observation that individual GAT3 blockade increased oscillation periods (*Figure 1E*). In contrast, when *sim*Dual-Block parameters were applied in the network, oscillations did not arise, consistent with the experimental observation that dual GAT1+GAT3 blockade inevitably abolished oscillations

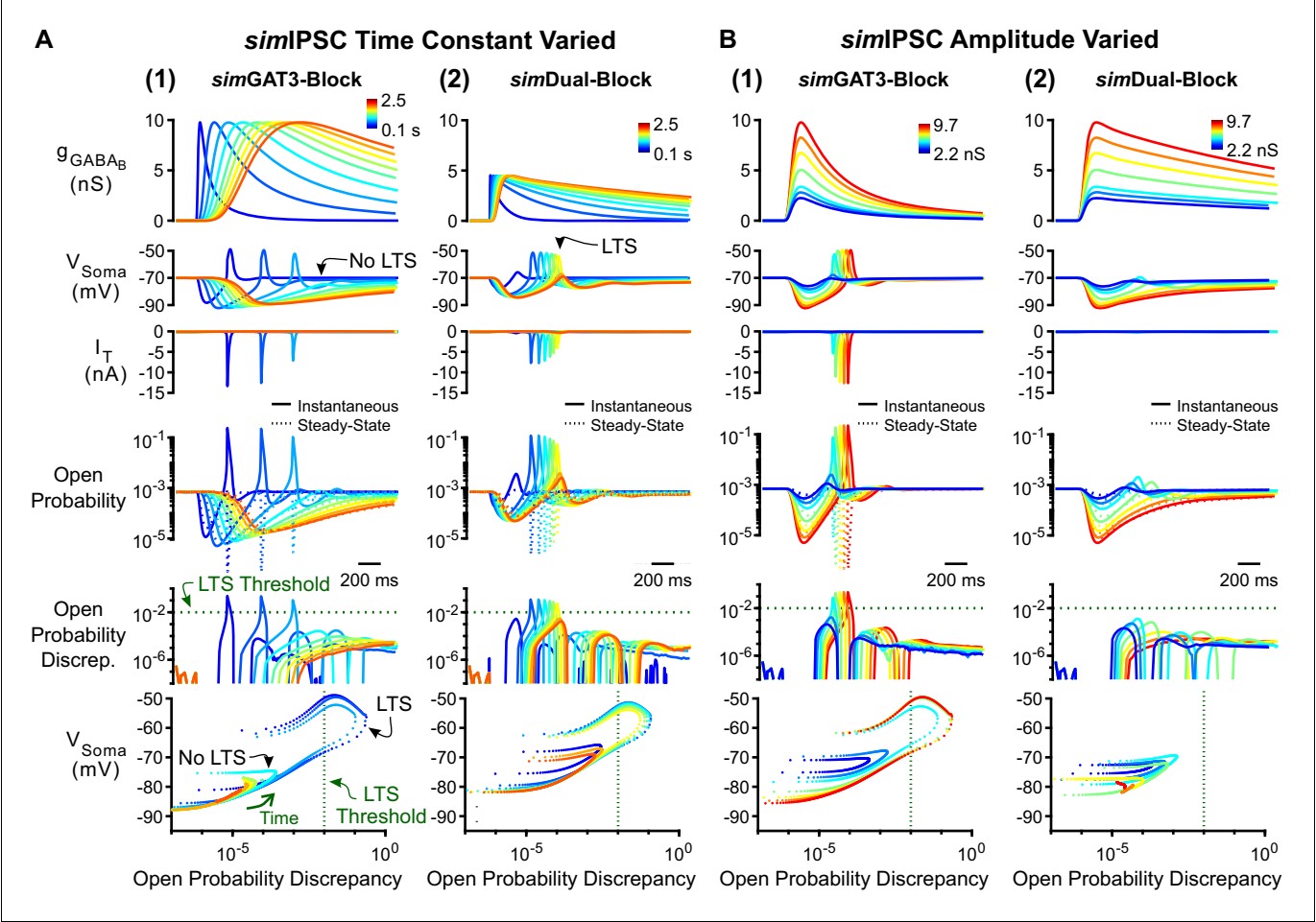

**Figure 7.** Manipulation of GABA_B IPSC kinetics bidirectionally modulates low-threshold rebound spike production. (**A**) (**1**) LTS responses to the *sim*GAT3-Block waveform (blue) gradually disappeared as the time constant was increased (with amplitude fixed) to that of the *sim*Dual-Block waveform (red). The green dotted line denotes the observed LTS threshold from *Figure 5C*. (**2**) LTS responses to the *s*Dual-Block waveform (red) gradually appeared as the time constant of the waveform was decreased (with amplitude fixed) to that of the *s*GAT3-Block waveform (blue). (**B**) (**1**) LTS responses gradually appeared as the amplitude of the *sim*GAT3-Block waveform was increased (with time constant fixed). (**2**) Robust LTS responses never appeared as the amplitude of the *sim*Dual-Block waveform is increased (with time constant fixed). This panel provides an explanation for the stark contrast between *dyn*GAT3-Block and *dyn*Dual-Block in *Figure 2E*.

The online version of this article includes the following figure supplement(s) for figure 7:

**Figure supplement 1.** Manipulation of GABA_B IPSC kinetics controlled for overall level of inhibition modulates low-threshold rebound spike production.

(*Figure 1D and F*). Additionally, across all two-cell networks, we found that oscillations only appeared when T channel open probability discrepancy passed the observed LTS threshold of $10^{-2}$ identified earlier (*Figure 8—figure supplement 1B*, *Figure 5C*). Indeed, T channel open probability discrepancy correlated well with oscillation probability (*Figure 8E*) and underscores the importance of the interplay between GABA_B-mediated inhibition and T channel gating in regulating oscillations. Although the two-cell networks recapitulated several effects of GAT blockade on oscillations, the oscillations generated by each two-cell network were nonetheless extremely stereotyped and regular, resulting in unrealistically high oscillatory indices (*Figure 8—figure supplement 1A*). Moreover, network responses across all two-cell networks were highly heterogeneous, with many of them failing to oscillate at the conductance amplitude scale used.

We sought to determine whether larger, more complex model networks could more realistically simulate experimental oscillations and recapitulate the bidirectional effects of GAT blockade by varying *sim*IPSCs. We scaled up the network to include one circular layer of 100 reticular thalamic (RT)

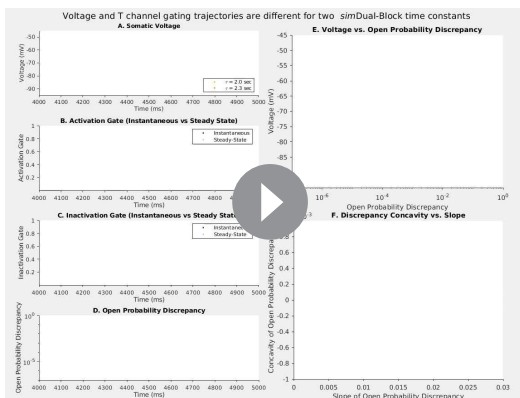

**Video 3.** Voltage and T-type calcium channel open probability discrepancy trajectories in response to GABA$_B$ IPSCs with two different time constants. Simulated traces for the example neuron of *Figure 3*, resampled at 1 ms intervals, in response to GABA$_B$ IPSCs using the *sim*Dual-Block waveform, but with time constants set to 2.0 (yellow) or 2.3 (orange) s. These two time constants correspond to the last LTS success and the first LTS failure, respectively, in *Figure 7A2*. (A-F) See descriptions for *Video 1*.
https://elifesciences.org/articles/59548#video3

neurons and one circular layer of 100 thalamo-cortical (TC) neurons (*Figure 9A*). RT-TC inhibitory connections and TC-RT excitatory connections were both convergent and divergent. To assess the importance of the geometric and conductance heterogeneity we observed in the single cell models (*Figure 3F*), we established two sets of model thalamic networks: (1) 24 *TC-homogeneous* networks with TC parameters taken one at a time from each of the 24 model neurons used in the two-cell networks and (2) 24 *TC-heterogeneous* networks with TC parameters taken from all of the 24 model neurons, randomly ordered. All networks had a quiescent, pre-stimulation baseline when the thalamocortical neuron leak reversal potential was varied between −73 and −60 mV (14 leak reversal potentials). To generate trial-to-trial variability (five trials per leak reversal potential), we randomized the leak conductance for each of the 200 neurons to within 10% of the original value. For both *TC-homogeneous* and *TC-heterogeneous* networks, oscillations emerged and spread in response to some but not all GABA$_B$ receptor activation parameters (*Figure 9B and C*). An oscillation probability was computed for each 200 cell network over the 14 × 5 = 70 trials. In addition, an oscillatory period, an oscillatory index and a half activation time was computed for each successfully-evoked oscillation (see Materials and methods).

We examined the distributions of oscillation measures over the set of *TC-homogeneous* networks (*Figure 9D*) and the set of *TC-heterogeneous* networks (*Figure 9E*). The values of oscillation periods and oscillatory indices for TC-heterogenous networks were similar to values extracted from experimental recordings (oscillation period: *Figure 1E*, oscillatory index: not shown). In response to the same *sim*IPSC conditions, we observed highly varied (often bi-modal) oscillation responses across TC-homogeneous networks, which reflects the highly varied LTS responses across individual model TC neurons (*Figure 4A*). In contrast, oscillation measures were less variable across the different TC-heterogeneous networks. Therefore, cell heterogeneity averages out the LTS response variability, provides more robust network responses and amplifies the differences across *sim*IPSC conditions. Indeed, for the set of *TC-heterogeneous* networks, relative to using *sim*Control parameters, oscillation probability increased when using either *sim*GAT1-Block (+8.2%, n = 24 networks, p=0.049) or *sim*GAT3-Block parameters (+8.5%, p=0.013), but decreased when using *sim*Dual-Block parameters (−82%, p=0.0010). Consistent with this result, the percent of active TC cells increased when using either *sim*GAT1-Block (+135%, p=0.0044) or *sim*GAT3-Block

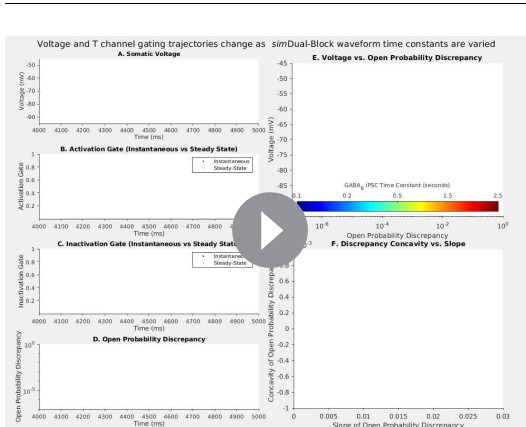

**Video 4.** Voltage and T-type calcium channel open probability discrepancy trajectories in response to GABA$_B$ IPSCs with varying time constants. Same as *Video 3* but in response to all GABA$_B$ IPSC time constants shown in *Figure 7A2*. (A-F) See descriptions for *Video 1*. There appears to be a threshold for the slope of the open probability discrepancy curve that differentiates between an LTS-producing trajectory and an LTS-lacking trajectory.
https://elifesciences.org/articles/59548#video4

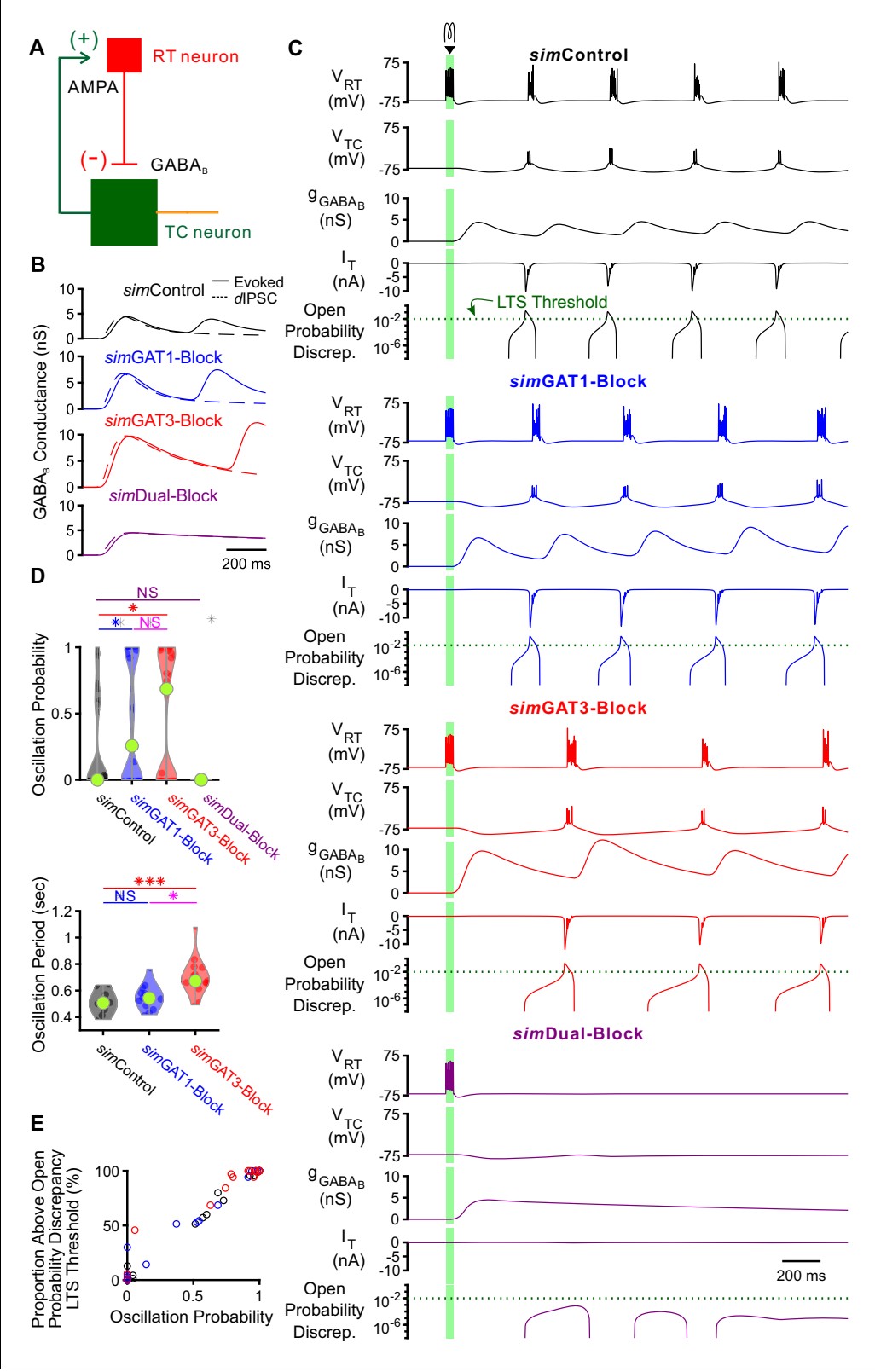

**Figure 8.** GABA$_B$-receptor mediated conductance waveforms modulate oscillations produced by two-cell model thalamic networks. (**A**) Schematic of a two-cell model network. A reticular thalamic (RT) neuron projected a GABA$_B$ receptor-mediated inhibitory synapse (-) to a thalamocortical (TC) neuron, which reciprocally projected an AMPA receptor-mediated excitatory synapse (+) to the reticular thalamic neuron. (**B**) Evoked GABA$_B$ conductance

*Figure 8 continued on next page*

*Figure 8 continued*

waveforms in network model TC neurons (solid lines) were similar to GABA$_B$ *dyn*IPSC waveforms (dashed lines). (**C**) Example two-cell network responses under different GABA$_B$ receptor conditions. Model TC neuron parameters corresponded to the example neuron of *Figure 3*. A brief (40 ms, 0.2 nA) current stimulus was applied to the reticular thalamic neuron, evoking an initial burst of 12 spikes. Oscillations were evoked under some but not all GABA$_B$ receptor conditions. A total of 24 model TC neurons produced oscillations in response to stimulation. The green dotted line denotes the observed LTS threshold from *Figure 5C*. (**D**) Distributions of oscillation measures over all 24, two-cell networks. Oscillation probability was increased when either *sim*GAT1-Block parameters or *sim*GAT3-Block parameters were used, but decreased when *sim*Dual-Block parameters were used (\*p<0.05, \*\*\*p<0.001, Friedman's test). (**E**) Proportion of simulations with the TC neuron's maximum T channel open probability discrepancy between 2 and 3 s after stimulation passing the LTS threshold shown in (**C**) correlated well with oscillation probability. Each circle denotes a different two-cell network using a specific *sim*IPSC color-coded as in (**C**). Responses across two-cell networks were heterogeneous.

The online version of this article includes the following source data and figure supplement(s) for figure 8:

**Source data 1.** Oscillation measures for two-cell model thalamic networks using different GABA$_B$ receptor activation waveforms (*sim*IPSCs).

**Figure supplement 1.** Additional information on two-cell model thalamic networks.

---

parameters (+193%, p=1.6×10$^{-5}$), but decreased when using *sim*Dual-Block parameters (−79%, p=0.037). Finally, average oscillation period increased when using either *sim*GAT3-Block (+23%, p=1.3×10$^{-7}$) or *sim*Dual-Block parameters (+50%, p=3.8×10$^{-9}$). These results are consistent with experimental findings that individual GAT1 or GAT3 blockade increased oscillation durations (*Figure 1D*) and oscillation periods (*Figure 1E*), whereas dual GAT1+GAT3 blockade eliminated oscillations (*Figure 1F*) in acute thalamic slices.

In summary, a population of thalamic network models was established that sufficiently recapitulated the bidirectional effects of individual versus dual GAT blockade on thalamic oscillations by merely altering the kinetics of GABA$_B$-receptor inhibition. The same interplay between GABA$_B$-receptor-mediated inhibition and T channel open probability that governs thalamocortical neuron rebound bursting appears to regulate network-level oscillations. Furthermore, we found that including cell heterogeneity in network simulations provides more robust oscillations and more realistically recapitulates experimental oscillation periods by averaging out LTS response heterogeneity. Using heterogeneous networks in future studies would thus facilitate comparison across experimental conditions and provide more meaningful predictions.

## Discussion

Here, we show that seizure-like thalamic oscillations were prolonged following individual GAT1 or GAT3 blockade, yet abolished following dual GAT1+GAT3 blockade. We have also shown that relative to control GABA$_B$ IPSC waveform responses, thalamocortical neuron rebound burst probability increased following individual GAT1 or GAT3 blockade waveforms, but decreased following dual GAT1+GAT3 blockade waveforms. These rebound burst effects were recapitulated in a large population of computational neurons modeled after biological thalamocortical neurons. When constructed into model thalamic networks, the computational neurons also produced oscillations that responded to different GAT blockade conditions in a manner similar to those recorded in situ. We found that including heterogeneity across thalamocortical neurons in the network significantly increased the robustness of evoked oscillations and accentuated the observed differences across GAT blockade conditions. Finally, we have identified and characterized a link between GABA$_B$-mediated inhibition and T channel gating across both voltage and time dimensions. Specifically, we found that the discrepancy between the instantaneous and the steady-state T channel open probability follows one of two trajectories: (1) failing to reach a threshold, resulting in LTS failure, or (2) driving past the threshold to produce a rebound burst. We discovered that inhibition during single GAT1 or GAT3 blockade waveforms could follow trajectory (2) as the amplitude is increased, but inhibition during the dual GAT1+GAT3 blockade waveform always followed trajectory (1) regardless of amplitude. These observations provide an explanation for the bidirectional effects of GAT blockade on both thalamocortical rebound bursting and seizure-like oscillations.

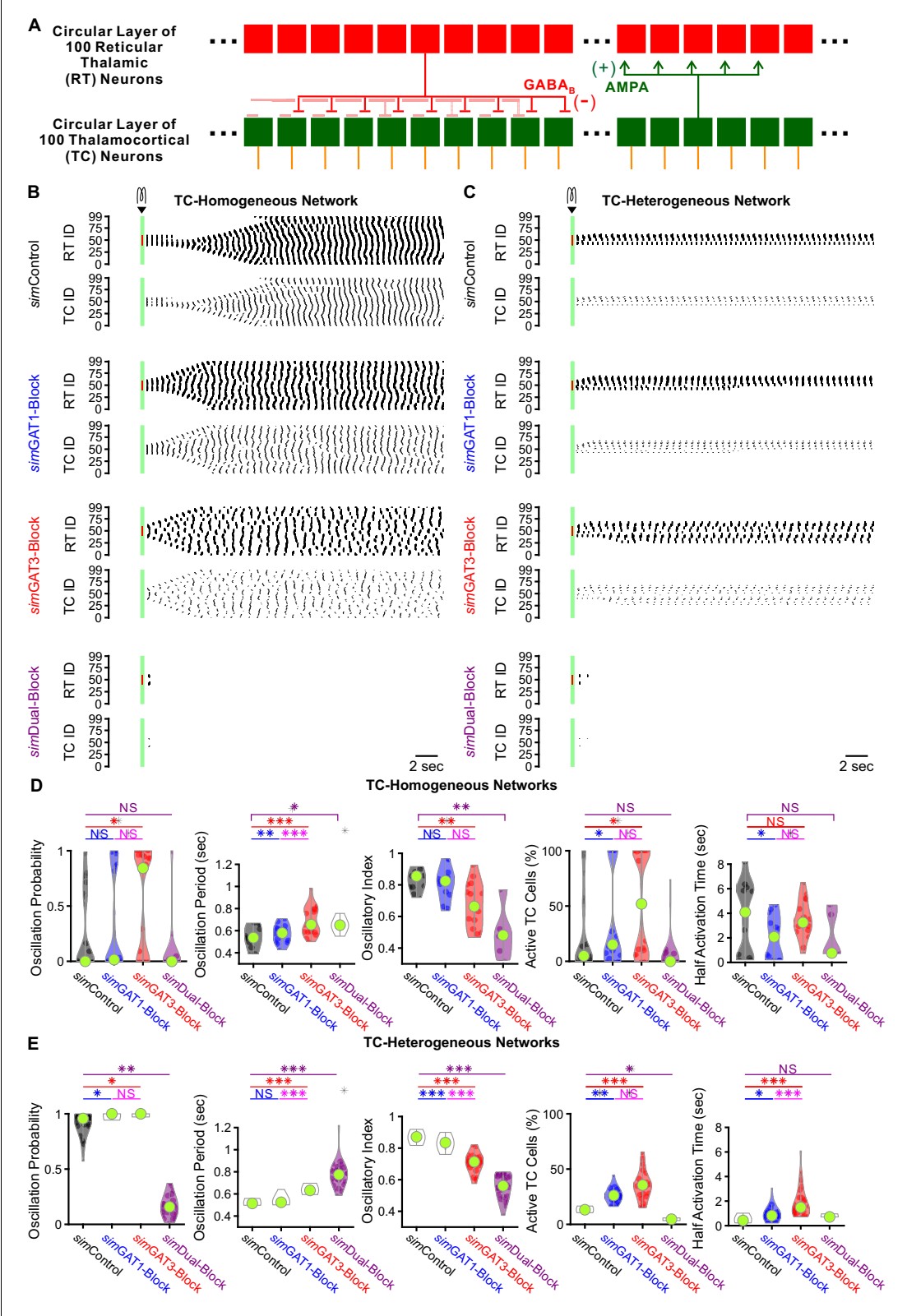

**Figure 9.** GABA$_B$-receptor-mediated conductance waveforms modulate oscillations produced by 200 cell model thalamic networks. (**A**) Schematic of a 200 cell model network. Each reticular thalamic (RT) neuron projected GABA$_B$ receptor-mediated inhibitory synapses (-) to nine nearby thalamocortical (TC) neurons. Each TC neuron projected AMPA-receptor-mediated synapses (+) to five nearby reticular thalamic neurons. (**B**) Sample spike raster plots of a TC-homogenous network, using model TC neuron parameters corresponding to the example neuron of *Figure 3*. A brief (40 ms, 0.2 nA) current

*Figure 9 continued on next page*

*Figure 9 continued*
stimulus was applied to each of the center 20 reticular thalamic neurons. Spikes within the stimulation period are red; all other evoked spikes are black. (C) Sample spike raster plots of a TC-heterogeneous network, using model TC neuron parameters corresponding to the 24 model TC neurons used in *Figure 8*. Relative to TC-homogeneous networks, activity was more localized for TC-heterogeneous networks. (D) Distributions of oscillation measures over all 24 TC-homogeneous 200 cell networks (*p<0.05, **p<0.01, ***p<0.001, repeated-measures ANOVA for oscillation period and half activation time, Friedman's test otherwise). (E) Distributions of oscillation measures over 24 randomly ordered, TC-heterogeneous 200 cell networks. Oscillation probability, oscillation period and percent of active cells increased when *sim*GAT3-Block parameters were used, but decreased when *sim*Dual-Block parameters were used (repeated-measures ANOVA for oscillatory index, Friedman's test otherwise).

The online version of this article includes the following source data for figure 9:

**Source data 1.** Oscillation measures for 200 cell model thalamic networks using different GABA$_B$ receptor activation waveforms (*sim*IPSCs).

## Role of thalamocortical neuron rebound bursting in generalized seizures

Thalamocortical neuron rebound bursting has long been implicated in spike-wave discharges (SWDs) observed in generalized seizures, and T channels mediate thalamocortical neuron rebound bursting (*Kim et al., 2001*; *Porcello et al., 2003*). The expression of the CaV3.1 T channel subtype by thalamocortical neurons correlates with SWD expression in both animal models (*Kim et al., 2001*; *Broicher et al., 2008*; *Ernst et al., 2009*) and human patients (*Singh et al., 2007*). Blocking T channels reduces both thalamocortical neuron bursting and oscillations in thalamic slice models (*Huguenard and Prince, 1994*). However, the importance of thalamocortical neuron bursting in generalized seizures remains unresolved. One recent study found reduced thalamocortical neuron firing during SWDs and a lack of seizure reduction with weak, local T channel blockade in the ventrobasal nucleus (*McCafferty et al., 2018*; however, strong blockade diminished SWDs), while a second recent study found that increasing/decreasing thalamocortical neuron bursting increases/decreases SWDs in both epileptic mice and rats (*Sorokin et al., 2017*). One possibility accounting for these discrepant findings is that the population of active thalamocortical neurons during SWDs is sparse (*Huguenard, 2019*). Interestingly, our TC-heterogenous network models – models that are likely more representative of biological networks – were largely characterized by robust, yet sparse, oscillations (in contrast, TC-homogenous network models produced oscillations that were widespread, *Figure 9B* with *Figure 9C*).

Our study found a strong relationship between thalamocortical neuron bursting and epileptiform thalamic oscillations. *First*, the pharmacological conditions in which oscillation *durations* were increased (individual GAT1 or GAT3 blockade) or decreased (dual GAT1+GAT3 blockade) relative to baseline were the same conditions in which thalamocortical neuron rebound burst *probability* increased (individual GAT1- or GAT3-Block GABA$_B$ *dyn*IPSC waveform) or decreased (Dual-Block GABA$_B$ *dyn*IPSC waveform), relative to control *dyn*IPSCs. *Second*, the pharmacological conditions in which oscillation *periods* were increased (individual GAT1 or GAT3 blockade) relative to baseline were also the same conditions in which thalamocortical neuron rebound burst *latencies* increased (individual GAT1- or GAT3-Block *dyn*IPSC waveform) relative to control *dyn*IPSCs. *Finally*, in our two-cell model thalamic network, each successive oscillation cycle was initiated by a thalamocortical neuron rebound burst (*Figure 8C*), similar to what has been reported in experiments when a reticular thalamic neuron and a thalamocortical neuron were simultaneously recorded during a thalamic oscillation (*Bal et al., 1995*).

## Role of the inhibitory temporal envelope in seizures

Prior experimental and computational work has suggested that a shift from GABA$_A$ receptor-mediated to GABA$_B$ receptor-mediated inhibition at the RT-TC synapse transforms oscillations in acute thalamic slices from a 10 Hz, sparse, spindle-like activity to a 3 Hz, hyper-synchronized, seizure-like state (*von Krosigk et al., 1993*; *Destexhe et al., 1996*; *Destexhe, 1998*; *Blumenfeld and McCormick, 2000*). Notably, the shift in oscillation frequency is consistent with differences in IPSC decay constants (GABA$_A$: <100 ms; GABA$_B$: about 300 ms) recorded in thalamocortical neurons (*Huguenard and Prince, 1994*). Multiple animal model studies support the hypothesis that generalized spike-wave seizures rely on robust GABA$_B$ receptor-mediated inhibition. Specifically, systemic injection of GABA$_B$ receptor agonists increases SWDs (*Liu et al., 1992*; *Bortolato et al., 2010*), while injection of GABA$_B$ receptor antagonists reduces or even abolishes SWDs (*Liu et al., 1992*;

*Vergnes et al., 1997*). Notably, however, other studies were not able to record rhythmic $GABA_B$ IPSCs during SWDs in vivo (*Charpier et al., 1999*) or find significant SWD changes with $GABA_B$ receptor modulation (*Staak and Pape, 2001*), and have highlighted a particular role for extrasynaptic $GABA_A$ receptors that produce a non-dynamic, tonic current in thalamocortical neurons (*Cope et al., 2009*). Presumably, tonic $GABA_A$ currents hyperpolarize the resting membrane potential, promoting T channel recovery and increasing rebound bursting (*Cope et al., 2005*). Thus, the disparate conclusions regarding the importance of $GABA_A$ receptor- versus $GABA_B$-receptor-mediated inhibition appear to nonetheless converge on similar conclusions regarding the importance of T channel recovery, a process that involves membrane potential hyperpolarization. However, we show here that hyperpolarization must be dynamic for T channels to produce an LTS. Under physiological conditions, it seems reasonable to expect multiple, convergent inhibitory mechanisms that promote T channel recovery.

Our study establishes a clear link between $GABA_B$ receptor-mediated inhibition, thalamocortical neuron rebound bursting and epileptiform thalamic oscillations. *First*, we found a 1.4-fold or a two-fold increase in oscillation duration in acute thalamic slices after perfusing with the GAT1 blocker NO-711 or the GAT3 blocker SNAP-5114 (*Figure 1D*), closely corresponding with the 1.5-fold or 2.2-fold increase in $GABA_B$ IPSC amplitude recorded under the same conditions, respectively (*Beenhakker and Huguenard, 2010*). *Second*, in both dynamic clamp recordings and model neuron simulations, the GAT1-Block and GAT3-Block $GABA_B$ IPSC waveforms produced higher thalamocortical neuron rebound LTS or burst probability relative to the Control waveform, with the GAT3-Block waveform increasing LTS or burst probability more (*Figure 2D* and *Figure 4A–B*). *Third*, we found that increasing conductances for each of the Control, GAT1-Block and GAT3-Block waveforms led to an increase in LTS or burst probability, for both dynamic clamp recordings and model neuron simulations (*Figure 2E* and *Figure 4C–D*). *Finally*, a comparison of the Control (black) versus GAT1-Block (blue) IPSC responses in *Figure 5* shows that the lack of LTS in the former correlates with a decreased level of TC hyperpolarization, a lack of T channel de-inactivation (both $h_T$ and $h_{\infty,T}$ are low) and a deficiency in T current production (no spike in $I_T$), agreeing with prior studies that show how the initiation of a low-threshold rebound spike depends on the sufficient removal of T channel inactivation through membrane potential hyperpolarization (*Llinás and Jahnsen, 1982*; *Coulter et al., 1989*).

More interestingly, we found that not only is the overall amount of inhibition important, how such inhibition distributes over time is equally important. For instance, the *dyn*Dual-Block waveform has an area under the curve (i.e. $\approx$ charge) about twice that of the *dyn*GAT3-Block waveform (*Figure 2B*). Nevertheless, both dynamic clamp recordings and model neuron simulations showed that the *dyn*Dual-Block waveform decreased rebound burst probability relative to the control, whereas the *dyn*GAT3-Block waveform increased rebound burst probability (*Figure 2D* and *Figure 4*). In fact, the *dyn*Dual-Block waveform largely failed to produce rebound bursts even when the conductance amplitude was scaled so high that the burst probability was close to one in all other conditions, that is following *dyn*Control, *dyn*GAT1-Block and *dyn*GAT3-Block waveforms scaled at 400%+ (*Figure 2E* and *Figure 4*). Dual GAT blockade also abolished oscillations, in stark contrast to the robust prolongation of oscillations observed during GAT3 blockade only (*Figure 1D and F*). Therefore, even high levels of synaptic inhibition, if decayed too slowly, can abolish both thalamocortical neuron rebound bursts and epileptiform thalamic oscillations, a conclusion recapitulated by our model thalamic networks (*Figure 8D* and *9E*).

## T channel open probability discrepancy drives thalamocortical neuron rebound bursting

Our model neurons allowed us to identify a novel measure of T channel gating that provides an explanation for how GAT-modulated, $GABA_B$ IPSCs bidirectionally regulates LTS production. We found that when the discrepancy between high instantaneous T channel open probabilities ($m_T^2 h_T$) and low steady-state open probabilities ($m_{T,\infty}^2 h_{T,\infty}$) passed a threshold of $10^{-2}$, thalamocortical neurons produced an LTS (*Figure 5C*) and two-cell thalamic networks produced oscillations (*Figure 8— figure supplement 1B*). To reach LTS threshold, we identified two criteria that must occur in the open probability discrepancy curve: (1) reach an inflection point and (2) the inflection point must have a steep slope. We compared three qualitatively different IPSC response trajectories in *Figure 5*

and *Video 2*. We observed that *sim*Control (black) did not produce an LTS response simply because inhibition was insufficient for T channel recovery ($h_T$ was always below 0.2, *Figure 5B*). Under these circumstances, the open probability discrepancy curve never reached zero concavity [failed criterion (1)]. Although T channels were sufficiently recovered ($h_T$ reached above 0.6) by strong hyperpolarization associated with *sim*GAT1-, *sim*GAT3-, and *sim*Dual-Block, only the former two waveforms produced an LTS response. Rapid depolarization during the decay of *sim*GAT1- and *sim*GAT3-Block (blue and red) caused the open probability discrepancy curve to not only reach an inflection point [satisfied criterion (1)], but also had a steeper slope and progressed towards positive concavity [satisfied criterion (2)], ultimately driving the open probability past LTS threshold. In contrast, the *sim*Dual-Block waveform (purple) produced a prolonged inhibition. As a result of the slower depolarization during IPSC decay, the open probability discrepancy curve did reach an inflection point [satisfied criterion (1)], but had a shallower slope at that inflection point [failed criterion (2)], causing the T channel open probability discrepancy to veer away from LTS threshold.

We conclude that a combination of high inhibition amplitude and fast inhibition kinetics is necessary for LTS production. A high inhibition amplitude causes sufficient T channel recovery, which appears to be necessary for the open probability discrepancy curve to reach an inflection point [criterion (1) above]. Fast inhibition kinetics causes rapid depolarization during IPSC decay, which appears to be necessary for the slope of the open probability discrepancy curve to be high at the inflection point [criterion (2) above], eventually driving the open probability discrepancy past LTS threshold. This open probability discrepancy depends on a slow T channel inactivation time constant, as doubling the T channel inactivation time constant produces LTSs in response to the *sim*Dual-Block waveform (*Figure 6B*), a waveform that normally results in LTS failure. In order to confirm that inhibition amplitude and kinetics influence LTS production by controlling T channel opening, we varied the GABA$_B$ *sim*IPSC waveform systematically five different ways (*Figure 7A–B*, *Figure 7—figure supplement 1*). In each case, an LTS was only produced when there was both sufficient T channel recovery and an increase in T channel open probability discrepancy. Collectively, the results of these manipulations (*Figure 7B*, *Video 3* and *Video 4*) support the hypothesis that dual GAT1+GAT3 blockade eliminates thalamic oscillations by promoting sustained, GABA$_B$ receptor-mediated inhibition that, in turn, promotes a convergence of instantaneous and steady-state T channel open probabilities, a condition that results in LTS failure.

## Modeling thalamocortical neuron heterogeneity

Prior experimental work has shown that T channels are critical for the generation of post-inhibitory low-threshold rebound spikes in thalamocortical neurons (*Kim et al., 2001*; *Porcello et al., 2003*). A computational study by *Destexhe et al., 1998* showed that membrane voltage trajectories of low-threshold rebound spikes in thalamocortical neurons can be recapitulated using a three-compartment model, but not a single-compartment model. A recent study by *Amarillo et al., 2014* used pharmacological approaches to identify seven intrinsic channels ($I_T$, $I_h$, $I_A$, $I_{Kir}$, $I_{NaP}$, $I_{NaLeak}$, $I_{KLeak}$) contributing to the resting membrane potential of thalamocortical neurons. The same study created a single-compartment computational model neuron that produced thalamocortical neuron rebound bursting and showed that a balance between the five voltage-dependent channels shapes the low-threshold rebound spike.

Our model thalamocortical neurons extend the studies by *Destexhe et al., 1998* and *Amarillo et al., 2014*. That is, we incorporated the same intrinsic channels described by *Amarillo et al., 2014*, but allowed channel density to vary across three compartments [note that the two different leak channels in Amarillo et al. were simplified by combining them into a single passive leak conductance ($g_{pas}$) with a reversal potential ($E_{pas}$)]. Furthermore, we established a different model neuron for each set of single thalamocortical neuron recordings (*Figure 3*), in an effort to capture the heterogeneity in response to physiologically relevant inhibitory inputs such as GABA$_B$ receptor-mediated IPSCs (*Figure 2*). By creating a total of 31 well-fitted model neurons, we were also able to generate population statistics that were in many ways similar to those of the corresponding population of recorded neurons (*Figure 4*). Having many distinct model thalamocortical neurons enabled us to introduce TC neuron heterogeneity into thalamic networks, which we found essential for robustly reproducing oscillations and capturing the differences across GAT blockade conditions seen in our slice recordings (*Figure 9*). Future work may utilize the distinct sets of optimized

parameters (*Figure 3F*) to probe for mechanisms underlying the high IPSC response heterogeneity seen in thalamocortical neurons in response to the same IPSCs (*Figure 2D*).

## Implication on anti-epileptic therapies

Drugs that increase synaptic inhibition in the brain may paradoxically exacerbate generalized seizures. For example, the GAT1 blocker tiagabine increases extracellular GABA concentrations (*Beenhakker and Huguenard, 2010*) and is an effective anti-epileptic drug used to treat focal epilepsy (*Nielsen et al., 1991*; *Uthman et al., 1998*). However, tiagabine can also induce continuous SWDs (i.e. absence status epilepticus) in patients with absence epilepsy (*Knake et al., 1999*) and non-convulsive status epilepticus in patients with focal epilepsy (*Ettinger et al., 1999*; *Vinton et al., 2005*). GAT1 knockout mice, which have increased inhibitory currents, and presumed increased activation of GABA$_B$ receptors (*Beenhakker and Huguenard, 2010*) in thalamocortical neurons, also exhibit increased incidence of SWDs (*Cope et al., 2009*). Furthermore, application of GABA$_B$ receptor antagonists increases seizures in rats susceptible to convulsive focal seizures, but suppresses seizures in rats with non-convulsive absence seizures (*Vergnes et al., 1997*). Together, these observations suggest that, although a seizure is a manifestation of hyperexcitable neuronal firing, increasing inhibition in the form of neuronal hyperpolarization may not always reduce seizures (*Beenhakker and Huguenard, 2009*).

In fact, as we discovered in this study, inhibition kinetics seems to play an important role in determining whether thalamocortical rebound bursts and seizure-like oscillations occur. The finding that dual GAT1+GAT3 blockade abolished epileptiform oscillations may at first glance suggest that non-specific GABA transporter blockers may be used to treat generalized seizures. However, this approach would be relatively non-specific and, for example, likely disrupt normal spindle oscillation formation, leading to undesirable side effects such as sedation and coma. Nevertheless, it is possible that temporary alteration of GABA$_B$ inhibition kinetics, either through GABA transporter modulation or other means, may create pockets of IPSC response heterogeneity in the overall thalamocortical neuron population, making it less likely for the thalamic network to develop a hypersynchronous state (*Pita-Almenar et al., 2014*).

## Materials and methods

### Oscillation recordings in acute thalamic slices

Sprague-Dawley rats [Charles River Laboratories (RRID:RGD_734476), Wilmington, MA] of postnatal day 11 to 17 (P11-P17), of either sex, were used in oscillation experiments, which were performed in accordance with protocols approved by the Institutional Animal Care and Use Committee at the University of Virginia (Charlottesville, VA). Rats were deeply anesthetized with pentobarbital, then transcardially perfused with ice-cold protective recovery solution containing the following (in mM): 92 NMDG, 25 glucose, 5 Na-ascorbate, 3 Na-pyruvate, 2.5 KCl, two thiourea, 20 HEPES, 26 NaHCO$_3$, 1.25 NaH$_2$PO$_4$, 0.5 CaCl$_2$, 10 MgSO$_4$, titrated to a pH of 7.3–7.4 with HCl (*Ting et al., 2014*). Horizontal slices (400 µm) containing the thalamus were cut in ice-cold protective recovery solution using a vibratome (VT1200, Leica Biosystems, Wetzlar, Germany). Slices were trimmed to remove the hippocampus and the hypothalamus, and then transferred to protective recovery solution maintained at 32–34°C for 12 min. Brain slices were kept in room temperature ACSF consisting of the following (in mM): 126 NaCl, 2.5 KCl, 10 glucose, 26 NaHCO$_3$, 1.25 NaH$_2$PO$_4$, 2 CaCl$_2$, and 1 MgSO$_4$. All solutions were equilibrated with 95% O$_2$/5% CO$_2$.

Slices were placed in a humidified, oxygenated interface recording chamber and perfused with oxygenated ACSF (2 mL/min) at 32–34°C. 10 µM bicuculline was added to the ACSF (bicuculline-ACSF) to block GABA$_A$ receptors. Oscillations were evoked by a square voltage pulse (10 V, 0.5 ms duration) delivered once every 60 s through two parallel tungsten electrodes (50–100 kΩ, FHC) 50–100 µm apart and placed in either the internal capsule or the reticular thalamus, which stimulated traversing corticothalamic and thalamocortical axons. Extracellular potentials were recorded in a differential manner with two tungsten electrodes (50–100 kΩ, FHC) by placing one in the somatosensory ventrobasal nuclei of the thalamus close to the stimulating electrode and one far away from the stimulating electrode. One experiment was performed per slice. Multi-unit recordings were amplified 10,000 times with a P511 AC amplifier (Grass), digitized at 10 kHz with Digidata 1440A, band-

pass filtered between 100 Hz and 3 kHz, and acquired using Clampex 10.7 software (Molecular Devices, San Jose, CA).

After at least 20 min in baseline bicuculline-ACSF, slices were perfused with one of 4 possible solutions: (1) bicuculline-ACSF, (2) bicuculline-ACSF plus 4 µM NO-711 to block GAT1 transport (*Sitte et al., 2002*), (3) bicuculline-ACSF plus 100 µM SNAP-5114 to block GAT3 transport (*Borden et al., 1994*) or (4) bicuculline-ACSF plus a combination of 4 µM NO-711 and 100 µM SNAP-5114 to simultaneously block both GAT1 and GAT3. After 40 min, the perfusion solution was switched back to bicuculline-ACSF for at least 20 min.

## Dynamic clamp recordings

Male Sprague-Dawley rats of postnatal day 11 to 15 (P11-P15) were used in dynamic clamp experiments, which were performed in accordance with protocols approved by the Administrative Panel on Laboratory Animal Care at Stanford University (Palo Alto, CA). Rats were deeply anesthetized with pentobarbital, then brains were rapidly extracted and placed in ice-cold protective recovery solution containing the following (in mM): 34 sucrose, 2.5 KCl, 11 glucose, 26 NaHCO$_3$, 1.25 NaH$_2$PO$_4$, 0.5 CaCl$_2$, and 10 MgSO$_4$, titrated to a pH of 7.4 with HCl. Horizontal slices (300 µm) containing the thalamus were cut in ice-cold protective recovery solution using a vibratome [VT1200 (RRID:SCR_018453), Leica Biosystems]. Slices were transferred to artificial cerebrospinal fluid (ACSF) maintained at 32°C for 45–60 min, then gradually brought to room temperature. The ACSF contained the following (in mM): 126 NaCl, 2.5 KCl, 10 glucose, 26 NaHCO$_3$, 1.25 NaH$_2$PO$_4$, 2 CaCl$_2$, and 1 MgSO$_4$. All solutions were equilibrated with 95% O$_2$/5% CO$_2$.

Slices were placed in a submerged recording chamber and perfused with oxygenated ACSF (2 mL/min) at 32–34°C. The chamber contained nylon netting which suspended the slice 1–2 mm from the chamber floor and enhanced slice perfusion. Slices were visualized with Dodt-contrast optics (Luigs and Newmann, Ratingen, Germany) on an Axioskop microscope (Zeiss, Pleasanton, CA). Recordings were obtained with a MultiClamp 700A patch amplifier (RRID:SCR_011323, Molecular Devices), digitized with Digidata 1322A and acquired using Clampex software. Borosilicate glass pipettes (1.5–3 MΩ) pulled on a P-87 micropipette puller (Sutter Instruments, Novato, CA) were filled with an internal solution containing (in mM): 100 potassium gluconate, 13 KCl, 10 EGTA, 10 HEPES, 9 MgCl$_2$, 2 Na$_2$-ATP, 0.5 Na-GTP, 0.07 CaCl$_2$ (pH 7.4).

Dynamic clamp experiments were conducted using a computer running the RealTime Application Interface for Linux (RTAI, www.rtai.org) sampling the intracellular potential at 50 kHz. Custom-written software modified from previous work (*Sohal et al., 2006*) used the sampled membrane potential and the pre-specified GABA$_B$-mediated inhibitory conductance to, at every timestep, update the inhibitory current that was injected into the thalamocortical neuron at 50 kHz. All measured voltages were corrected for a junction potential of −10 mV.

Inhibitory conductance waveforms (*dyn*Control, *dyn*GAT1-Block, *dyn*GAT3-Block, *dyn*Dual-Block, *Figure 2B*) for use in dynamic clamp experiments were created based on previous recordings of GABA$_B$-mediated IPSCs (*Beenhakker and Huguenard, 2010*). In the previous study, voltage-clamped thalamocortical neurons were recorded during electrical stimulation of presynaptic reticular thalamic neurons while the thalamic slices were bathed in ionotropic glutamate receptor and GABA$_A$ antagonists to isolate GABA$_B$-mediated IPSCs. After acquiring baseline data, one of three drugs was added to the perfusing ACSF: 4 µM NO-711 to block GAT1, 100 µM SNAP-5114 to block GAT3, or both 4 µM NO-711 and 100 µM SNAP-5114 to block both GAT1 and GAT3 (*Borden et al., 1994*; *Sitte et al., 2002*). After at least 10 min of drug perfusion and stabilization of drug effect, a series of GABA$_B$-mediated IPSCs was acquired. In the present study, we converted the previously obtained averaged IPSCs for each drug condition into a conductance waveform fitted with the equation:

$$g_{GABA_B} = A\left(1 - e^{-t/\tau_{rise}}\right)^8 \left(we^{-t/\tau_{fallfast}} + (1-w)e^{-t/\tau_{fallslow}}\right) \tag{1}$$

where $A$ is the amplitude coefficient, $\tau_{rise}$ is the rise time constant, $\tau_{fallfast}$ and $\tau_{fallslow}$ are the fast and slow decay constants, respectively, and $w$ is the weighting factor for the two decay terms (*Otis et al., 1993*). The percent change in each parameter from baseline to drug condition was computed for each experiment. Baseline-normalized parameter values were compared between conditions, and for parameters with significant population differences in values (p<0.05, Wilcoxon signed rank test), the mean change in parameter value was found and implemented for that drug condition.

For parameters that did not change significantly in the drug condition compared to baseline, the parameter value was kept identical to that of the baseline (*dyn*Control) condition for the templates (*Figure 2B*). Parameter values for the four experiment conditions (*dyn*Control, *dyn*GAT1-Block, *dyn*GAT3-Block, *dyn*Dual-Block) are listed in *Table 2*. A value of -115 mV was used for the reversal potential associated with the GIRK conductance following post-synaptic GABA$_B$ receptor activation.

Since the number of GABA$_B$ receptors activated in a physiological setting was not determined, all GABA$_B$ IPSC waveforms (*dyn*IPSCs) had amplitude scaled between 25–800%. A total of 5–15 repetitions were performed for each *dyn*IPSC waveform, with a holding current adjusted at the beginning of each recording so the holding membrane potential is in the range of $-73$ to $-60$ mV. For each sweep, a brief current pulse ($-50$ pA, 10 ms) was applied at around 100 ms to assess electrode resistance and passive membrane properties, then the *dyn*IPSC was applied at 1000 ms.

## Analysis of oscillation recordings

MATLAB R2018a (RRID:SCR_001622, MathWorks, Natick, MA) was used for all data analysis. Single action potential *spikes* were detected from raw multi-unit activity as follows (*Sohal et al., 2003*). The raw signal was first bandpass-filtered between 100 and 1000 Hz. Slopes between consecutive sample points were computed from the filtered signal. For each sweep $i$, the baseline slope noise $\alpha_i$ was computed from the root-mean-square average of the slope vector over the baseline region before stimulation start, and the maximum slope $\beta_i$ was defined as the maximum slope value at least 25 ms after stimulation start (to account for the stimulus artifact). For each slice, if $\alpha = \sum_i \alpha_i$ is the baseline slope noise averaged over all sweeps and $\beta = \sum_i \beta_i$ is the maximum slope averaged over all sweeps, then the slice-dependent signal-to-noise ratio was given by $r = 1 + 0.1\left(\frac{\beta}{\alpha} - 1\right)$. The resulting signal-to-noise ratios fell between 2 and 5. A sweep-dependent slope threshold was then defined by $\theta_i = r\alpha_i$. Finally, single spikes were defined as all local maxima of the slope vector at least 25 ms after stimulation start with values exceeding a slope threshold.

To compute the oscillation duration for each sweep, spikes were first binned by 10 ms intervals to yield a spike histogram. Evoked *bursts* were detected by joining consecutive bins with a minimum spike rate of 100 Hz, using a minimum burst length of 60 ms, a maximum delay after stimulation start of 2000 ms and a maximum inter-burst interval of 2000 ms (*Figure 1A*). The *oscillation duration* was defined as the time difference between the end of the last evoked burst and stimulation start.

To compute the oscillation period for each sweep, an autocorrelation function (ACF) was first computed from the binned spikes, then moving-average-filtered using a 100 ms window. Peaks (local maxima) were detected from the filtered ACF using a minimum peak prominence that is 0.02 of the amplitude of the primary peak (the first value of the ACF). Peaks with lags greater than the oscillation duration were ignored. The *oscillation period* was computed by searching for the lag value $\delta$ that minimizes the distance function $f(\delta) = \sum_j |p_j - q_j(\delta)|$, where $p_j$ is the lag value of peak $j$ and $q_j(\delta)$ is the closest multiple of $\delta$ to $p_j$. The search was initialized with the estimate $\delta_0 = p_2 - p_1$ and confined to the bounds $\left[\frac{2}{3}\delta_0, \frac{3}{2}\delta_0\right]$.

An *oscillatory index* (*Sohal et al., 2006*) was also computed from the filtered autocorrelation function (fACF). For each sweep, the non-primary peak with largest fACF value was designated as the secondary peak. The *oscillatory index* was then defined by $OI = \left(A_{peak2} - A_{trough}\right)/\left(A_{peak1} - A_{trough}\right)$, where $A_{peak1}$ is the fACF value of the primary peak, $A_{peak2}$ is the fACF value of the secondary peak and $A_{trough}$ is the minimum fACF value between the primary peak and the secondary peak.

The average oscillation duration or period at the end of each phase (baseline or drug) was computed by choosing the five values from the last 10 sweeps of each phase that were within 40% of the average of the group of values. This approach was used to minimize bias caused by abnormally-shortened evoked oscillations due to the presence of spontaneous oscillations.

## Analysis of dynamic clamp recordings

MATLAB R2018a was used for all data analyses. Responses to *dyn*IPSCs were analyzed as follows: All voltage traces were manually examined and noisy recordings were excluded. The peak of the current trace (IPSC peak) within the first 300 ms of *dyn*IPSC start was first detected. The most likely candidate for a calcium-dependent low-threshold spike (LTS) was then detected from the raw voltage trace in between the time of IPSC peak and 7000 ms after *dyn*IPSC start.

**Table 2.** Parameter values of GABA$_B$-mediated inhibitory conductance waveforms used in dynamic clamp experiments.

These values (with the amplitudes scaled by 200%) correspond to the conductance templates shown in **Figure 2B**. A, amplitude coefficient; $\tau_{rise}$, rise time constant; $\tau_{fallfast}$, fast decay time constant; $\tau_{fallslow}$, slow decay time constant; $w$, weighting factor for the two decay terms.

| | A | $\tau_{rise}$ | $\tau_{fallfast}$ | $\tau_{fallslow}$ | $w$ |
|---|---|---|---|---|---|
| *dyn*Control | 16.00 | 52.00 | 90.10 | 1073.20 | 0.952 |
| *dyn*GAT1-Block | 24.00 | 52.00 | 90.10 | 1073.20 | 0.952 |
| *dyn*GAT3-Block | 8.88 | 38.63 | 273.40 | 1022.00 | 0.775 |
| *dyn*Dual-Block | 6.32 | 39.88 | 65.80 | 2600.00 | 0.629 |

To detect the *LTS candidate*, the voltage trace was first median-filtered with a time window of 30 ms to remove action potentials (*Chung et al., 2002*), then moving-average-filtered with a time window of 30 ms. First and second derivatives of the doubly-filtered voltage traces were computed by differences between consecutive sample points, and the first derivative vector was moving-average-filtered with a time window of 30 ms before taking the second derivative. The local maximum of the doubly-filtered voltage trace with the most negative second derivative was chosen as the LTS candidate. A histogram of LTS candidate second derivatives for traces was fitted to a sum of 3 Gaussian distributions and the minimum of the probability density function between the first two peaks was used as a second derivative threshold.

Features were computed for each LTS candidate. The LTS *peak value* was the absolute voltage value for the peak. The LTS *latency* was defined by the time difference between the LTS peak time and *dyn*IPSC start. A time region that was bounded by the first local minimum of the doubly-filtered voltage trace on either side of the peak was used to compute the LTS *maximum slope* and detect action potentials. For accuracy, the slope value was computed from a different doubly filtered trace with a smaller moving-average-filter time window corresponding to roughly a 3 mV-change in amplitude for that trace. Action potentials spikes were detected by a relative amplitude threshold 10 mV above the LTS amplitude for that trace.

An LTS candidate was then assigned as an *LTS* if the following criteria were all satisfied: (1) Peak prominence must be greater than the standard deviation of the filtered voltage values before IPSC start; (2) Peak second derivative must be more negative than the threshold ($-2.3$ V/s$^2$) described above; (3) If there were action potentials riding on the LTS, the LTS peak time must occur after the time of the first action potential. Detection results were manually examined by blinded experts and the algorithm's decision for LTS determination was overturned only if three of the four polled electrophysiology experts agreed. The *LTS or burst probability* was defined as the proportion of traces producing an LTS or a *burst* (an LTS with at least one riding action potential) across all 5–15 repetitions (with varying holding potentials) for a particular GABA$_B$ IPSC waveform.

## Single thalamocortical neuron models

All computational simulations were performed using NEURON (RRID:SCR_005393, *Carnevale and Hines, 2009*) version 7.5 with a temperature of 33°C. Each model thalamocortical neuron had one cylindrical somatic compartment (Soma) and two cylindrical dendritic compartments (Dend1 and Dend2) in series (*Figure 3C*). The somatic length and diameter were set to be equivalent with the parameter $diam_{soma}$. The two dendritic compartments were set to have equal diameters ($diam_{dend}$), with each having lengths equal to half of the parameter $L_{dend}$. Values for specific membrane capacitance and axial resistivity were equivalent to those reported by *Destexhe et al., 1998*. Passive leak channels were inserted in all three compartments at equivalent densities ($g_{pas}$) with a fixed reversal potential of -70 mV. The 4 voltage-independent (*passive*) parameters described above were allowed to vary across neurons.

The following mechanisms were inserted in all three compartments: the T-type calcium current ($I_T$) and the submembranal calcium extrusion mechanism ($Ca_{decay}$) was adapted from *Destexhe et al., 1998*; the hyperpolarization-activated nonspecific cationic current ($I_h$), the A-type transient potassium current ($I_A$), the inward-rectifying potassium current ($I_{Kir}$) and the persistent sodium current

($I_{NaP}$) were adapted from *Amarillo et al., 2014*. The following parameters were allowed to vary across compartments and across neurons: the maximum permeability $\bar{p}_T$ (in cm/s) of $I_T$, which was described by the Goldman–Hodgkin–Katz flux equation (*Huguenard and McCormick, 1992*), and the maximum conductance densities $\bar{g}_h$, $\bar{g}_A$, $\bar{g}_{Kir}$, $\bar{g}_{NaP}$ (in S/cm$^2$) of other currents described by Ohm's law. All parameters that were not varied during optimization were identical for all model neurons and taken from literature values. Based on calculations from solutions used in experiments, the potassium reversal potential was -100 mV, the sodium reversal potential was 88 mV, the calcium concentration outside neurons was 2 mM and the initial calcium concentration inside neurons was 240 nM.

All intrinsic current mechanisms were described by Hodgkin-Huxley type equations with voltage and/or time-dependent activation ($m$) and inactivation ($h$) gating variables and have been described in detail in the corresponding sources (*Amarillo et al., 2014*; *Destexhe et al., 1998*). In particular, the open probability of $I_T$ was described by $m(V,t)^2 h(V,t)$ (*Huguenard and Prince, 1992*). At every point in time, the activation variable $m$ converges exponentially to its steady-state value $m_\infty(V)$ with a time constant $\tau_m(V)$ and the inactivation variable $h$ converges exponentially to its steady-state value $h_\infty(V)$ with a time constant $\tau_h(V)$. The equations for $m_\infty(V)$, $\tau_m(V)$, $h_\infty(V)$ and $\tau_h(V)$ were fitted from electrophysiological recordings and described by *Huguenard and McCormick, 1992*.

For simulations involving action potentials, a fast sodium and potassium mechanism adapted from *Sohal and Huguenard, 2003* was inserted in the somatic compartment and made identical across neurons. Based on the comparison of the number of spikes per LTS between model neurons and corresponding recorded neurons (*Figure 4—figure supplement 1A–B*), the threshold parameter $V_{Traub}$ was changed to $-65$ mV.

A custom GABA$_B$ receptor mechanism was inserted in the somatic compartment that produced IPSCs with the equation form given by *Equation 1*. The parameters used were identical to that used for dynamic clamp, with kinetics varying for each pharmacological condition as in *Table 2*. Although parameter measurements were performed at a same temperature (33°C) as simulations, a Q$_{10}$ of 2.1 (*Otis et al., 1993*) was included in the model.

All simulations performed for each model neuron were matched to traces recorded using dynamic clamp for the corresponding recorded neuron as follows: To match the holding potentials, we first performed a test simulation to voltage clamp the model neuron (with initial potential $-70$ mV) for 2 s to bring it to a quasi-steady state. The resultant steady-state current was used as holding for the current clamp simulation. To simulate dynamic clamp of the recorded neuron, after 2 s to allow state variables in the model to stabilize, a brief current pulse ($-50$ pA, 10 ms) was applied at around 2.1 s, then an *sim*IPSC identical to the *dyn*IPSC applied in dynamic clamp was applied at 3 s. The analysis of LTS and burst features for simulated *sim*IPSC responses is identical to that for recorded *dyn*IPSC responses.

## Parameter initialization

Initial values for *passive parameters* were different for each model neuron and were estimated from the current pulse responses from the corresponding recorded neuron by a strategy adopted from *Johnston and Wu, 1994*. A portion of raw current pulse responses had a systematic voltage shift at the beginning and end of the pulse, consistent with an unbalanced bridge. These shifts were detected by a slope threshold determined through a histogram of all initial slopes and then corrected by shifting the entire portion of the response during the brief current pulse, by the calculated amount. The corrected current pulse responses for a particular neuron were then fitted to the following first order response equation with two exponential components:

$$V(t) = \left[C_0\left(1 - e^{-t/\tau_0}\right) + C_1\left(1 - e^{-t/\tau_1}\right)\right]B(t \leq t_p) + \left[C_0\left(1 - e^{-t_p/\tau_0}\right)e^{-(t-t_p)/\tau_0} + C_1\left(1 - e^{-t_p/\tau_1}\right)e^{-(t-t_p)/\tau_1}\right]B(t > t_p)$$

where $C_0$, $C_1$ are the amplitudes (in mV) of the two components, $\tau_0$, $\tau_1$ are the time constants (in ms) of the two components, $t_p = 10\,\text{ms}$ is the width of the current pulse, and $B(x)$ is a Boolean function defined by $B(x) = 1$ if $x = \text{true}$ and $B(x) = 0$ if $x = \text{false}$. Initial values for the curve fit were $C_{0,i} = V$, $C_{1,i} = V$, $\tau_{0,i} = 10\,\text{ms}$, $\tau_{1,i} = 1\,\text{ms}$ where $V$ is the mean voltage change for the neuron after each 10 ms stimulus.

Next, given the fitted coefficients, we estimated the defining parameters for a ball-and-stick model. The input resistance was computed by

$$R_{Input} = (C_0 + C_1)/I_p,$$

where $I_p = 50\ \text{pA}$ is the amplitude of the current pulse. The membrane time constant $\tau_m$ was set to equal that of the slow component $\tau_m = \tau_0$. The length constant was computed by solving for $L$ in equation 4.5.57 of Johnston and Wu:

$$|C_1/((2C_0\tau_1/\tau_0) - C_1)| = \cot(\alpha_1 L)[\cot(\alpha_1 L) - 1/\alpha_1 L],$$

where $\alpha_1 = ((\tau_1/\tau_0) - 1)^{1/2}$, starting with the initial guess of $L_i = \pi/\alpha_1$. The dendritic-to-somatic conductance ratio $\rho$ was then computed by equation 4.5.58 of Johnston and Wu:

$$\rho = -\alpha_1 \cot(\alpha_1 L)/\coth(L).$$

Finally, we converted the defining ball-and-stick parameters to initial estimates for the passive parameters in our 3-compartment NEURON model that are optimized. The input resistances of the soma and the dendrite were computed by

$$R_{Memb} = R_{Input} - R_s, R_{Soma} = (1 + \rho)R_{Memb}, R_{Dend} = R_{Soma}/\rho,$$

where $R_s$ is the series resistance. The specific membrane resistivity $R_m$ was computed by equation 4.3.3 of Johnston and Wu:

$$R_m = \tau_m/C_m,$$

where a fixed value of 0.88 μF/cm$^2$ (*Destexhe et al., 1998*) for the specific membrane capacitance $C_m$. Then the somatic diameter $diam_{Soma}$ was computed by equation 4.3.8 of Johnston and Wu:

$$diam_{Soma} = 2\sqrt{R_m/4\pi R_{Soma}},$$

the dendritic diameter $diam_{Dend}$ was computed by equation 4.5.47 of Johnston and Wu:

$$diam_{Dend} = \left(2\sqrt{R_m R_a}(\coth L)/\pi R_{Dend}\right)^{2/3},$$

where a fixed value of 173 Ω·cm (*Destexhe et al., 1998*) for the axial resistivity $R_a$, the dendrite length $L_{Dend}$ was computed by equation 4.4.15 of Johnston and Wu:

$$L_{Dend} = L\sqrt{diam_{Dend} R_m/4R_a},$$

and the passive conductance $g_{pas}$ was estimated by

$$g_{pas} = 1/R_{Input}A,$$

where $A = \pi(diam_{Soma}^2 + diam_{Dend}L_{Dend})$ is the total surface area of the neuron.

Initial values for *active parameters* were identical for all model neurons and were taken from literature values (*Amarillo et al., 2014*; *Destexhe, 1998*) as follows: $\bar{p}_{T,Soma} = \bar{p}_{T,Dend1} = \bar{p}_{T,Dend2} = 2.0\ \text{x}\ 10^{-4}\ \text{cm/s}$, $\bar{g}_{h,Soma} = \bar{g}_{h,Dend1} = \bar{g}_{h,Dend2} = 2.2\ \text{x}\ 10^{-5}\ \text{S/cm}^2$, $\bar{g}_{Kir,Soma} = \bar{g}_{Kir,Dend1} = \bar{g}_{Kir,Dend2} = 2.0\ \text{x}\ 10^{-5}\ \text{S/cm}^2$, $\bar{g}_{A,Soma} = \bar{g}_{A,Dend1} = \bar{g}_{A,Dend2} = 5.5\ \text{x}\ 10^{-3}\ \text{S/cm}^2$ and $\bar{g}_{NaP,Soma} = \bar{g}_{NaP,Dend1} = \bar{g}_{NaP,Dend2} = 5.5\ \text{x}\ 10^{-6}\ \text{S/cm}^2$.

## Parameter optimization

The voltage responses of a model neuron to a set of 12 different GABA$_B$ IPSC waveforms (four pharmacological conditions at three different conductance amplitude scales) was compared with the responses of the corresponding recorded neuron. To emphasize the LTS response, model neurons excluded fast sodium and potassium channels and the experimental responses were median-filtered with a time window of 30 ms to remove action potentials. As multiple objective functions were of interest (*Druckmann et al., 2007*), a total error $E_{total}$ was computed by a weighted average of component errors (*Figure 3E*):

$$E_{total} = (w_m E_m + w_{sw} E_{sw} + w_a E_a + w_t E_t + w_{sl} E_{sl}) / \sum_i w_i,$$

where $E_m$ is the average error across all traces for whether the presence or absence of LTS matches (if an LTS is missed an error of 18 is assigned; if an LTS is falsely produced an error of 6 is assigned), $E_{sw}$ is the average root-mean-square error of voltage values, $E_a$ is the average LTS peak value error across all LTS-matching traces, $E_t$ is the average LTS latency error across all LTS-matching traces, $E_{sl}$ is the average LTS maximum slope error across all LTS-matching traces, and the $w_i$'s denote the corresponding weights. Based on the maximum LTS latency observed in recorded neurons, the fitting region was restricted to between 0 and 1.8 s after the start of the IPSC. Traces were also allowed to have different sweep weights, in which case the average across traces were computed using root-mean-square.

A modified version of the Nelder-Mead simplex algorithm (*Lagarias et al., 1998*) was used to update parameter values within defined parameter bounds, with the objective of minimizing $E_{total}$. For each simplex, the maximum number of iterations was 2000, the maximum number of error evaluations was 4000, the relative error tolerance was 0.1, the relative parameter change tolerance was 0.1, the coefficient of reflection was $\rho = 1$, the coefficient of expansion was $\chi = 2$, the coefficient of contraction was $\gamma = 0.75$, the coefficient of shrinkage was $\sigma = 0.8$. The initial simplex was set up so that each vertex deviates in a parameter to be varied by $\delta = 2/3$ of the parameter's total range. To transform values to an unconstrained parameter space so that the Nelder-Mead method could be applied, each bounded parameter value was first linearly transformed to the region $[-1, 1]$ than transformed to $(-\infty, \infty)$ using the inverse tangent function.

A total of 21 different iterations were run to optimize each model neuron, with the weights among different error types and weights among different traces varying from iteration to iteration. For some poorly fit neurons, the initial parameters for that neuron were replaced by the best-fit neuron's parameters from the previous iteration. One trace per GABA_B IPSC waveform was used during model optimization, but all of 60–180 traces were used for the final model evaluation (*Figure 3E*).

## Network models

Each 2 cell network includes one single-compartment model reticular thalamic neuron, described in previous work (*Klein et al., 2018*), and one of the well-fitted, three-compartment model thalamocortical neurons described earlier. The reticular thalamic neuron has an AMPA receptor that is synaptically activated by the thalamocortical neuron, and the thalamocortical neuron has a GABA_B receptor in the somatic compartment that is synaptically activated by the reticular thalamic neuron. Synaptic currents were evoked with 100% probability and a delay of 1 ms whenever the presynaptic neuron reached a voltage threshold of −30 mV.

AMPA receptors were adapted from *Sohal et al., 2000*. Reflecting more recent physiological measurements (*Deleuze and Huguenard, 2006*), AMPA currents were adjusted to bring rise time to 0.5 ms, decay time to 5.6 ms and maximal conductance of 7 nS per synapse. The reversal potential for AMPA currents was maintained at 0 mV.

GABA_B receptors were as described before for single model TC neurons. In order for overlapping IPSCs to be summed linearly, we expanded *Equation 1* into 18 terms, each having its own rise and decay exponential time constants. To be consistent with dynamic clamp experiments, a reversal potential of −115 mV was used in simulations, although the networks behave similarly if a reversal potential of −100 mV was used instead (data not shown). Relative to the values used in dynamic clamp, the conductance amplitudes were scaled by 1/12 to account for temporal summation from an RT burst, and the synaptically-evoked IPSCs was verified to be comparable to recorded values (*Figure 8B*).

Each 200 cell network assumed a bilayer architecture, with one 100 cell circular layer of single-compartment model reticular thalamic neurons and one 100 cell circular layer of three-compartment model TC neurons. The identity of the TC neuron was chosen from each of the 31 well-fitted model TC neurons. Each reticular thalamic neuron inhibited the nine nearest TC cells, whereas each TC neuron excited the five nearest reticular thalamic neurons (*Sohal and Huguenard, 2003*). The passive leak conductance of each model reticular thalamic neuron was randomly selected from a uniform distribution between 45 and 55 μS/cm$^2$ (*Sohal and Huguenard, 2003*). The passive leak conductance

of each model TC neuron was randomly selected from a uniform distribution between $-10$ and 10% of the optimized value.

Network simulations were performed with leak reversal potentials and initial membrane potentials set to a value between $-73$ mV and $-60$ mV, at 1 mV increments, so that a total of 14 repetitions were applied for each model network. After a delay of 3 s to allow for state variable stabilization, either the reticular thalamic neuron in the two-cell network or each of the center 20 reticular thalamic neurons in the 200 cell network was injected with a square current pulse (0.2 nA, 40 ms). Simulations were performed with a 0.1 ms integration time step and continued for a total of 30 s. Pooling all action potential spikes after stimulation end, we detected bursts and computed an *oscillation period* and an *oscillatory index* using the same algorithm as that for multiunit recordings, except for a bin width of 100 ms for spike histograms and a minimum peak prominence of 0.5 relative to the largest secondary peak for the filtered autocorrelation function. *Oscillation probability* was defined as the proportion of simulations (with varying holding potentials) that induced oscillations with at least three bursts. *Percent of active TC cells* was defined as the percentage of model TC neurons in the network that produced at least one spike. *Half activation latency* was defined as the time it took for half of the final percentage of active cells to be activated. Mean oscillation period, mean oscillatory index and mean half activation latency were computed by restricting to trials with a successfully evoked oscillation.

### Statistics

MATLAB R2019b was used for all statistical analysis. Since the number of available data points for LTS and burst features was significantly different for the Dual-Block condition, we performed a paired t-test or a signed-rank test between the Control condition and the Dual Block condition. We used either repeated-measures ANOVA or the Friedman's test for comparison across the Control, GAT1-Block and GAT3-Block conditions.

Paired comparisons were applied if not otherwise specified. Normality of the differences to the within-subject mean was assess for all groups using a combination of the Lilliefors test, the Anderson-Darling test and the Jarque-Bera test. Normality was satisfied when the geometric mean of three p values was at least 0.05 for all groups. When normality was satisfied, the paired-sample *t*-test was used when there are two groups and repeated-measures ANOVA (with multiple comparison) was used when there are more than two groups. When normality was not satisfied, the Wilcoxon signed-rank test was used when there are two groups and Friedman's test (with multiple comparison) was used when there are more than two groups. Tests were two-tailed with a significance level of 0.05. Error bars reflect 95% confidence intervals. All violin plots used a bandwidth that was 10% of the maximum data range.

### Drugs

Bicuculline methiodide was purchased from Sigma-Aldrich (St. Louis, MO). The GAT1 blocker NO-711 [(1,2,5,6-tetrahydro-1-[2-[[(diphenylmethylene)amino]oxy]ethyl]-3-pyridinecarboxylic acid hydrochloride)] and GAT3 blocker SNAP-5114 [1-[2-[tris(4-methoxyphenyl)methoxy]ethyl]-(S)-3-piperidinecarboxylic acid] were purchased from Tocris Bioscience (Minneapolis, MN).

### Code

Data analysis, data visualization, statistical tests, NEURON model setup and NEURON model optimization were performed using custom MATLAB code. All code for reproducing results (including NEURON. hoc,. tem and. mod files used) are available online at https://github.com/luadam4c/m3ha_published/. (*Lu, 2020*; copy archived at https://github.com/elifesciences-publications/m3ha_published).

## Acknowledgements

We thank the University of Virginia Medical Scientist Training Program, the Whitfield-Randolph Scholarship and the NIH grants R01-NS099586 and R01-NS034774 for funding support. We thank Shin-Shin Nien for help with figure organization. We thank Dr. Peter Klein, Dr. Lise Harbom and Katie Salvati for help with manual LTS scoring. We thank Dr. Farzan Nadim for providing guidance on computational modeling.

## Additional information

### Competing interests

John R Huguenard: Senior editor, *eLife*. The other authors declare that no competing interests exist.

### Funding

| Funder | Grant reference number | Author |
|---|---|---|
| National Institute of Neurological Disorders and Stroke | NIH grant R01-NS099586 | Adam C Lu<br>Brian Truong<br>Mark P Beenhakker |
| National Institute of Neurological Disorders and Stroke | NIH grant R01-NS034774 | Christine Kyuyoung Lee<br>Max Kleiman-Weiner<br>Megan Wang<br>John R Huguenard |
| University of Virginia | Whitfield-Randolph Scholarship | Adam C Lu |

The funders had no role in study design, data collection and interpretation, or the decision to submit the work for publication.

### Author contributions

Adam C Lu, Conceptualization, Data curation, Software, Formal analysis, Supervision, Validation, Investigation, Visualization, Methodology, Writing - original draft, Writing - review and editing; Christine Kyuyoung Lee, Conceptualization, Data curation, Software, Formal analysis, Investigation, Methodology, Writing - review and editing; Max Kleiman-Weiner, Software, Methodology, Writing - review and editing; Brian Truong, Software; Megan Wang, Methodology, Writing - review and editing; John R Huguenard, Conceptualization, Resources, Supervision, Funding acquisition, Methodology, Project administration, Writing - review and editing; Mark P Beenhakker, Conceptualization, Resources, Supervision, Funding acquisition, Investigation, Methodology, Project administration, Writing - review and editing

### Author ORCIDs

Adam C Lu https://orcid.org/0000-0002-1008-1057
Christine Kyuyoung Lee http://orcid.org/0000-0003-1422-4606
Max Kleiman-Weiner https://orcid.org/0000-0002-6067-3659
Brian Truong http://orcid.org/0000-0003-0179-0932
Megan Wang http://orcid.org/0000-0001-8845-4936
John R Huguenard https://orcid.org/0000-0002-6950-1191
Mark P Beenhakker https://orcid.org/0000-0002-4541-0201

### Ethics

Animal experimentation: Oscillation experiments were performed in accordance with Protocol #3892 approved by the Institutional Animal Care and Use Committee at the University of Virginia. Dynamic clamp experiments were performed in accordance with protocols approved by the Administrative Panel on Laboratory Animal Care at Stanford University. Rats were deeply anesthetized with pentobarbital before transcardial perfusion, and every effort was made to minimize suffering.

### Decision letter and Author response

Decision letter https://doi.org/10.7554/eLife.59548.sa1
Author response https://doi.org/10.7554/eLife.59548.sa2

## Additional files

### Supplementary files
• Transparent reporting form

### Data availability

Source data files have been provided for Figures 1-5, 8-9. Oscillations data, dynamic clamp data are available via Dryad (https://doi.org/10.5061/dryad.4xgxd256f). All code for reproducing results are available online at https://github.com/luadam4c/m3ha_published/ (copy archived at https://github.com/elifesciences-publications/m3ha_published).

The following dataset was generated:

| Author(s) | Year | Dataset title | Dataset URL | Database and Identifier |
|---|---|---|---|---|
| Beenhakker MP, Lu AC, Lee CK, Kleiman-Weiner M, Truong B, Wang M, Huguenard JR | 2020 | Nonlinearities between inhibition and T-type calcium channel activity bidirectionally regulate thalamic oscillations | https://doi.org/10.5061/dryad.4xgxd256f | Dryad Digital Repository, 10.5061/dryad.4xgxd256f |

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
