## [Decision Letter]

**Acceptance summary:**

This study effectively combines experimental and modeling work to show and explain the synergistic effects of blocking GABA transporters on thalamocortical oscillations. This is shown to be due to a dependence of T-type calcium channel gating on GABA_B_ receptor activity.

**Decision letter after peer review:**

Thank you for submitting your article "Nonlinearities between inhibition and T-type calcium channel activity bidirectionally regulate thalamic oscillations" for consideration by *eLife*. Your article has been reviewed by three peer reviewers, including Frances K Skinner as the Reviewing Editor and Reviewer #1, and the evaluation has been overseen by Ronald Calabrese as the Senior Editor. The following individual involved in review of your submission has agreed to reveal their identity: Maxim Bazhenov (Reviewer #2).

The reviewers have discussed the reviews with one another and the Reviewing Editor has drafted this decision to help you prepare a revised submission.

As the editors have judged that your manuscript is of interest, but as described below that additional information is required before it is published, we would like to draw your attention to changes in our revision policy that we have made in response to COVID-19 (https://elifesciences.org/articles/57162). First, because many researchers have temporarily lost access to the labs, we will give authors as much time as they need to submit revised manuscripts. We are also offering, if you choose, to post the manuscript to bioRxiv (if it is not already there) along with this decision letter and a formal designation that the manuscript is "in revision at *eLife*". Please let us know if you would like to pursue this option. (If your work is more suitable for medRxiv, you will need to post the preprint yourself, as the mechanisms for us to do so are still in development.)

Summary:

This paper addresses the ability of thalamic relay neurons to generate low threshold spikes (and bursts) under conditions when GABA-_A_-receptors are blocked, and a GABA-_B_ type conductance is applied. The GABA-_B_ conductance has amplitude and kinetics determined by the presence or absence, alone or together, of 2 types of GABA transporter (GAT1 and GAT3). The authors examine effects of GAT1, GAT3 and combined GAT1+GAT3 blocking on thalamocortical bursting activity using combination of experimental techniques and computational modeling (single cell and network). They report when GAT1 or GAT3 is blocked, there was an increase in the thalamocortical oscillations and bursting, however when both GAT 1 and 3 are blocked, the oscillations and bursting were abolished. Based on dynamic-clamp techniques and fitting to the model, the authors identified that interaction of GABA-_B_ with T-channel activation is critical for the differential effect. Specifically, the fast repolarization from a hyperpolarization state led to LTS and eventually to oscillations in case of GAT1/3 blocking. In contrast, blocking both GAT1 and 3 led to prolonged inhibition that prevented LTS. These finding suggests a careful balance between inhibition (GABA-_A_ and B) and calcium currents determines the LTS properties, and bursting in the thalamocortical neurons. Overall, solid work that combines experiment and modeling.

Essential revisions:

While all of the reviewers thought that this was very good work, they all struggled with presentation clarity and a clear summary message about mechanisms involved. The authors are asked to distill and improve the presentation of their work which is very heavy in places.

It was thought that if authors have a clear understanding of principles and mechanisms, they should be able to express it in a clear way in the Abstract/Introduction/Discussion.

For some of the points raised by the reviewers it was decided that it would be at the authors' discretion whether to include them as part of the present revisions (referred to as optional below). The authors could consider them in subsequent work.

1) Experiments: There seem to be washout effect differences with the 2 different blockers. This would seem to potentially play a role on the synergistic abolition? Did you examine oscillations for longer with GAT1 or GAT3 blockers separately as done for the combination? Also, based on the raw recordings shown in Figure 1(iv), the “elimination” could potentially be lower amplitude oscillations? I wondered whether spectral/frequency analyses would be helpful and/or show something additional to PSTHs.

2) Single cell experiments: The dynamic clamp GABA_B_ conductance “approach” would circumvent the experimental variability referred to, it does introduce a different issue in considering the biology – i.e., spatial aspects since one is only looking somatically. So, the comparison can be more “robust” but then the single neuron effect should perhaps be tempered? The authors should comment on how they expect that this would affect their results and interpretation. That is, the GAT3 blockage has an extended temporal effect which is further extended with the combination. I think this should be unpacked a bit more for interpretation purposes.

3) Once one sees Figure 2B, the rest of the aspects of the paper are pretty straightforward and then is mainly about heterogeneous responses moving forward. I am not sure if it is fair to so clearly link the thalamic oscillations directly to the rebound burst aspect (single cell) for these “bidirectional differences”. It then started to get a bit confusing about interpretation of it all (see above point). That is, I think that the authors are assuming an underlying mechanism about TC cells rebound and thalamic oscillations – presumably it is based on previous network modeling studies that they build on later – but the heterogeneous responses and 1 and 2 points above give me a bit of pause. At this stage, they have shown a correlation of network oscillations and single cell responses with GAT blockers.

4) Model: For the examples shown in Figure 3D, could the authors say which cases they are in 3E – presumably well-fit ones? I think that it would be helpful to show the non well-fit cases to see the difference (supplementary figures as in *eLife* structure could help with this presentation).

5) While the cellular heterogeneity modeling and importance is interesting, and the authors' use of 3 compartment models seems reasonable, the general statements about commonalities of T channel/A channel densities need to be tempered and/or explained a bit more, especially given the limited nature of the models. Figure 3F are where the various parameters are shown but hard to appreciate/understand them in light of the statements. A statement about negative correlation is mentioned as an example. The authors go on to fully explore T-type calcium channels, and bring about the temporal aspects which make sense given kinetic differences noted from the experimental work.

However, I did wonder what one would get if a thorough exploration was done with h-channels to understand the differential effects? Or was this clear already? (this latter point is optional to address in the revised manuscript).

6) The network oscillation modeling and heterogeneous aspects claimed seemed to leave out a lot without mention (see points 1 and 2 above for example). I was also not sure why and whether it was “fair” to add and consider the trial-to-trial variability via the leak reversal potential (given the different TC models)? For example, did it matter which TC model one chose (Figure 3E) to use for the homogeneous networks to compare with heterogeneous? Was some aspect from the 2-cell network used as guidance in the choice for the larger networks? (I may have missed that?)

7) Figure 1C: Why do oscillations seem much shorter "before" drug application in NO and SNAP conditions compare to Control? In fact, in NO condition oscillations seem to be lasting as long as in Control.

8) In both slice experiments and the model, the GABA-_A_ synapse was blocked by bicuculline. However, in-vivo condition would involve the interaction of both GABA-_A_ and GABA-_B_ channels. While this may be difficult to examine in the slice work, it is feasible to examine in the computational model. It would be important to understand how the GABA-_A_ fast acting time scale impacts the effect of slower changing GABA-_B_ synapse, especially in network simulations.

9) Based on the fitting of dynamic-clamp data to the model, the authors' predict high density of T-channel in dendrites and A-channel density in soma. While the rest of the work, especially the computational modeling, has focused on T-channel, the changes to A-channel have not been examined. Similar to the findings in Figure 5-6 on T-channel, it would be helpful to examine the effect of A-channel (this last point is optional to addressed in a revised manuscript).

10) Given that the T-channel is involved in the generation of spindle activity during sleep, it would be helpful to include some discussion about the impact of the findings on sleep spindles.

11) While the authors have developed an interesting measure called the T-channel open channel discrepancy, I feel that the effect could be well captured using conventional phase-plane or phase-space projections. It seems that the membrane voltage, T-channel activation and inactivation variables pass through a critical region in phase-space which is required for the LTS; the failure of this dynamics leads to the lack of LTS. Possibly 3D phase-space analysis of these variables could identify the critical region?

(this point is optional to address in a revised manuscript).

12) The section "Interplay between GABA_B_ receptors and T-type calcium channels " is extremely long and difficult to follow. I am sure some readers will follow all the details there but for many others the message may be lost. I suggest some rewriting to include summaries and highlight the main results.

13) In the network simulations (Figure 8), for TC-homogeneous case, oscillations seem to be more synchronized in control vs GAT1 or GAT3 block conditions. Why is this so?

14) The way the manuscript is written, it is somewhat difficult to learn what is actually a precise mechanism behind the differences between effects of individual GAT1/3 blocking vs dual blocking. E.g., the Summary in discussion says: "…respectively, can be recapitulated by varying the GABA_B_ receptor activation waveform." but it does not provide a needed summary of the mechanisms discovered in the study.

15) Consequences of the GABA-_B_ conductance properties are reflected in the behavior of individual cells, and of oscillating networks of TCR and nRT (nucleus reticularis) neurons. Details of the GABA-_B_ conductances to be used were determined in prior publications. Experimental and numerical experiments were performed in thalamocortical slices, in dynamic clamp applied to single TCR neurons, and in simulations of 3-compartment TCR neurons alone, paired with an nRT cell, and in a TCR/nRT network. The main finding is that excessively large and prolonged GABA-_B_ conductances interfere with LTS generation; detailed analysis shows that the specific kinetic properties of T-channels are responsible, in that LTS requires an epoch where inward current is regenerative (increasing m) at the same time as inactivation is not too developed, and the latter in turn depends on instantaneous inactivation (h) being larger than would occur at equilibrium. Such large slow conductances develop when both transport blockers are applied together, but not separately. The authors provide some discussion of possible relations to spike-wave seizures, and the effects and contrary side-effects of certain anticonvulsants.

I suspect that the mathematics involved is more general than discussed here for T-type calcium channels, and depends at heart (no pun intended) on the slowness of inactivation vs activation. An example might be the inability of a squid axon to fire at low frequencies, even in the presence of a low calcium buffer. The authors may, at their discretion, want to consider this larger issue (optional to address in a revised manuscript).

---

## [Author Response]

Essential revisions:While all of the reviewers thought that this was very good work, they all struggled with presentation clarity and a clear summary message about mechanisms involved. The authors are asked to distill and improve the presentation of their work which is very heavy in places.It was thought that if authors have a clear understanding of principles and mechanisms, they should be able to express it in a clear way in the Abstract/Introduction/Discussion.For some of the points raised by the reviewers it was decided that it would be at the authors' discretion whether to include them as part of the present revisions (referred to as optional below). The authors could consider them in subsequent work.1) Experiments: There seem to be washout effect differences with the 2 different blockers. This would seem to potentially play a role on the synergistic abolition? Did you examine oscillations for longer with GAT1 or GAT3 blockers separately as done for the combination? Also, based on the raw recordings shown in Figure 1(iv), the “elimination” could potentially be lower amplitude oscillations? I wondered whether spectral/frequency analyses would be helpful and/or show something additional to PSTHs.

The effects of washing out the drugs were not that different for NO-711 vs SNAP-5114 (during the washout of either drug, the duration of evoked oscillations decreased), although the washout was generally faster for NO-711. To prevent confusion, we have selected a more representative example for SNAP-5114. In Beenhakker and Huguenard, 2010, GABA_B_ IPSCs reached a stable plateau level for the duration of drug application within 10 minutes of applying either GAT1, GAT3, or dual GAT1+GAT3 blockers. Similarly, we observed in this study that oscillation durations stabilized within 15 minutes of applying either GAT1 or GAT3 blockers. We thus did not examine oscillations for longer than 40 minutes of either NO-711 or SNAP-5114 application because oscillations had stabilized and the prolongation in duration was robust (Figure 1D). Therefore, if wash-in periods were extended, we would not anticipate an eventual abolishment of oscillations following either single GAT1 or GAT3 blockade, unlike what we observed during dual GAT1+GAT3 blockade (Figure 1F).

We interpret “remnant low amplitude oscillations” as meaning the possibility of oscillatory spiking activity that is buried in the noise. The reviewer’s comment is well taken and we have now performed the requested spectral/frequency analysis for the recordings (Author response image 1). As we expected, the analysis failed to resolve any spiking activity in the recording of the dual block condition, while activity in the other conditions is readily apparent. Therefore, it seems quite unlikely that activity during the dual blockade condition is robust, but unresolvable with our spike detection algorithm. The original raw recordings in Figure 1B(iv) were selected for a slice where complete oscillation abolishment took longer than 40 minutes. To avoid further confusion, we have now selected a more representative example of a raw recording during dual blockade where complete oscillation abolishment occurred.

**Author response image 1. respfig1:** Individual GAT1 or GAT3 blockade strengthens thalamic oscillations, but dual GAT1+GAT3 blockade abolishes oscillations. Spectrograms for example evoked epileptiform oscillations at baseline and 40 minutes after perfusing with (i) control (no drug added), (ii) 4 µM NO-711 (GAT1 blocker), (iii) 100 µM SNAP-5114 (GAT3 blocker) or (iv) dual 4 µM NO-711+100 µM SNAP-5114. These are the same example oscillations as in Figure 1B. A bin width of 100 ms was used. Importantly, these analyses do not rely on any spike detection algorithms.

2) Single cell experiments: The dynamic clamp GABA_B_ conductance “approach” would circumvent the experimental variability referred to, it does introduce a different issue in considering the biology – i.e., spatial aspects since one is only looking somatically. So, the comparison can be more “robust” but then the single neuron effect should perhaps be tempered? The authors should comment on how they expect that this would affect their results and interpretation. That is, the GAT3 blockage has an extended temporal effect which is further extended with the combination. I think this should be unpacked a bit more for interpretation purposes.

Since the GABA_B_ receptor-mediated IPSCs in Beenhakker and Huguenard, 2010, were recorded from the soma, it is natural to introduce the corresponding conductance waveforms in the soma through dynamic clamp. Although this approach may lead to a delayed voltage response in the dendrites, the fact that voltage attenuation from soma to dendrite is small should alleviate this concern. In fact, our analysis of thalamocortical neuron current pulse responses yielded an average electrotonic length of 0.71 ± 0.08 space constants (range: 0.1-1.5 space constants). This means that voltage changes at the distal dendrite would be on average at least 50% of that in the soma at any given point in time. Furthermore, we saw the same rebound bursting differences across *dyn*IPSCs as the amplitude was scaled between a range of 50-800% in our dynamic clamp experiments (Figure 2E). These amplitude scales should address most voltage attenuation concerns. Nonetheless, we have included a cautionary statement in the text.

3) Once one sees Figure 2B, the rest of the aspects of the paper are pretty straightforward and then is mainly about heterogeneous responses moving forward. I am not sure if it is fair to so clearly link the thalamic oscillations directly to the rebound burst aspect (single cell) for these “bidirectional differences”. It then started to get a bit confusing about interpretation of it all (see above point). That is, I think that the authors are assuming an underlying mechanism about TC cells rebound and thalamic oscillations – presumably it is based on previous network modeling studies that they build on later – but the heterogeneous responses and 1 and 2 points above give me a bit of pause. At this stage, they have shown a correlation of network oscillations and single cell responses with GAT blockers.

We agree that we may have overstated the link between T channel gating and oscillations. The phrase "in agreement" has been changed to “correlated”. We have also cited studies that linked thalamocortical neuron rebound bursting with thalamic oscillations in the beginning of the "Single neuron recordings" section. Throughout the text, we have now clarified that the link between T channel gating and oscillations is at this point only correlational. We have also added a figure panel that shows a high correlation between our measure that predicts the production of a rebound low threshold spike in thalamocortical neurons (“open probability discrepancy”, Figure 5C) with the probability for oscillations to appear in a network (see Figure 8E).

4) Model: For the examples shown in Figure 3D, could the authors say which cases they are in 3E – presumably well-fit ones? I think that it would be helpful to show the non well-fit cases to see the difference (supplementary figures as in eLife structure could help with this presentation).

The ranks are now stated in the Figure 3 legend and labelled in Figure 3—figure supplement 1. We also provided fits for 4 additional well-fitted cases and the 2 excluded cases in the supplement.

5) While the cellular heterogeneity modeling and importance is interesting, and the authors' use of 3 compartment models seems reasonable, the general statements about commonalities of T channel/A channel densities need to be tempered and/or explained a bit more, especially given the limited nature of the models. Figure 3F are where the various parameters are shown but hard to appreciate/understand them in light of the statements. A statement about negative correlation is mentioned as an example. The authors go on to fully explore T-type calcium channels, and bring about the temporal aspects which make sense given kinetic differences noted from the experimental work.However, I did wonder what one would get if a thorough exploration was done with h-channels to understand the differential effects? Or was this clear already? (this latter point is optional to address in the revised manuscript).

We included general statements about commonalities of T channel/A channel densities because these findings converged on previous results in the literature. The relevant literature has now been cited where appropriate. Nonetheless, in order to focus the manuscript on the interplay between inhibition and T channels, we have now moved descriptions of other channels to Figure 5—figure supplement 1. We agree that a thorough exploration of h channels would be interesting, but it is probably beyond the scope of this work.

6) The network oscillation modeling and heterogeneous aspects claimed seemed to leave out a lot without mention (see points 1 and 2 above for example). I was also not sure why and whether it was “fair” to add and consider the trial-to-trial variability via the leak reversal potential (given the different TC models)? For example, did it matter which TC model one chose (Figure 3E) to use for the homogeneous networks to compare with heterogeneous? Was some aspect from the 2-cell network used as guidance in the choice for the larger networks? (I may have missed that?)

We apologize for the confusion. Leak reversal potential was not allowed to vary during the fitting process. All TC model neurons had a leak reversal potential of -70 mV. Therefore, we feel that it is fair to compare simulations derived from all 2-cell networks subjected to the same range of leak reversal potentials. We ran simulations using 14 different leak reversal potentials, and ran 5 different trials per leak reversal potential. The 5 trials differed by introducing a different leak conductance, and this leak conductance was randomized about a mean leak conductance that was based on the TC model for each network. It is customary to perform such randomization across trials (Sohal and Huguenard, 2003).

Each 2-cell network indeed behaved differently. To account for TC neuron heterogeneity, we examined all possible 2-cell networks using each of the well-fitted model TC neurons (Figure 8D). Similarly, we examined all possible TC-homogeneous networks using each of the well-fitted model TC neurons (Figure 9D). Due to the response heterogeneity, we also examined TC-heterogeneous networks that included all of the well-fitted model TC neurons, randomly ordered (Figure 9E). We now include a description of these analyses:

“In response to the same *sim*IPSC conditions, we observed highly varied (often bi-modal) oscillation responses across TC-homogeneous networks, which reflects the highly varied LTS responses across individual model TC neurons (Figure 4A). In contrast, oscillation measures were less variable across the different TC-heterogeneous networks. Therefore, cell heterogeneity averages out the LTS response variability, provides more robust network responses and amplifies the differences across *sim*IPSC conditions.”

7) Figure 1C: Why do oscillations seem much shorter "before" drug application in NO and SNAP conditions compare to Control? In fact, in NO condition oscillations seem to be lasting as long as in Control.

Oscillation durations are highly variable in acute slice experiments as the extent of the intact network in each slice is difficult to control. Examples shown in Figure 1C have been reselected to show similar baseline durations for Control, NO-711 and SNAP-5114. Furthermore, the distribution of baseline durations over all slices in Figure 1D were similar across all four pharmacological conditions (Control: 6.9 ± 2.4 seconds, NO-711: 6.6 ± 2.5 seconds, SNAP-5114: 4.9 ± 1.8 seconds, NO-711+ SNAP-5114: 7.7 ± 1.7 seconds).

8) In both slice experiments and the model, the GABA-_A_ synapse was blocked by bicuculline. However, in-vivo condition would involve the interaction of both GABA-_A_ and GABA-_B_ channels. While this may be difficult to examine in the slice work, it is feasible to examine in the computational model. It would be important to understand how the GABA-_A_ fast acting time scale impacts the effect of slower changing GABA-_B_ synapse, especially in network simulations.

We have now performed both 2-cell network simulations (Author response image 2) and 200-cell TC-heterogeneous network simulations (Author response image 3) with GABA_A_ receptors added to model TC neurons.

Observations on oscillations generated by networks (2 and 200) that incorporate GABA_A_ receptors include: (1) a decreased oscillation period across most conditions, as one would expect from the briefer GABA_A_ kinetics, and (2) a decreased oscillation probability when *sim*Control, *sim*GAT1-Block or *sim*GAT3-Block waveform was used. Unlike networks containing only GABA_B_ receptors (and unlike our experimental observations), oscillation probability increased when the *sim*Dual-Block waveform was used. We attribute this effect to the briefer decay kinetics provided by GABA_A_ receptor mediated inhibition, a condition that promotes rebound bursting by increasing T channel open probability discrepancy (as demonstrated in Figure 6D).

Nevertheless, the oscillation differences across GAT blockade conditions derived from network simulations including GABA_A_ receptors were the same as those derived from network simulations without GABA_A_ receptors (cf. Figure 9E and Author response image 3). That is, relative to using the *sim*Control waveform, we observed an increase in oscillation probability using the *sim*GAT1-Block or *sim*GAT3-Block waveform and a decrease in oscillation probability using the *sim*Dual-Block waveform. Relative to using the *sim*Control waveform, we also observed an increase in oscillation period using the *sim*GAT3-Block or *sim*Dual-Block waveform.

However, we have not included these results in the manuscript as we do not have either voltage clamp recordings to physiologically constrain GABA_A_ receptor parameters, or oscillation recordings to support the observations. In the simulations above, we simply applied GABA_A_ receptor conductance kinetics from Huntsman et al., 1999, and set an amplitude ratio of 2 (relative to the GABA_B_ conductance) estimated from Huguenard and Prince, 1994. However, these may not reflect physiological changes in GABA_A_ receptor mediated-IPSCs under various forms of GAT blockade. Furthermore, in order to support the observations above, one would need to record oscillations in which GABA_A_ receptors are present in TC neurons but not in RT neurons, which is not easy to do pharmacologically. One possibility is to either record oscillations in β3 subunit knockout mice (Huntsman et al., 1999) or a comparable genetically-modified rat, but we feel that is beyond the scope of this work. If the reviewers feel strongly about including these simulation findings, then we can add them as a supplement to Figure 9.

**Author response image 2. respfig2:** With GABA_A_ receptors present, GABA_B_-receptor mediated conductance waveforms still modulate oscillations produced by 2-cell model thalamic networks. (A) Schematic of a 2-cell model network with GABA_A_ receptors present (cf. Figure 8A). (B) Example 2-cell network responses under different GABA_B_ receptor conditions. The same TC model neuron and simulation conditions as that in Figure 8C were applied, except for the addition of a GABA_A_ conductance with maximal amplitude 2 times that of the maximal GABA_B_ conductance. The green dotted line denotes the observed LTS threshold from Figure 5C. (C) Distributions of oscillation measures over the same 24, 2-cell networks as in Figure 8D. Compared to Figure 8D, 2-cell network simulations with GABA_A_ receptors present had a lower oscillation probability and a smaller oscillation period (* p < 0.05, *** p < 0.001, Friedman’s test). The differences across simIPSCs were generally the same as in Figure 8D. (D) Figure 8D reproduced here for ease of comparison.

**Author response image 3. respfig3:** GABA_B_-receptor mediated conductance waveforms modulate oscillations produced by 200-cell model thalamic networks with GABA_A_ receptors present. (A) Schematic of a 200-cell model network with GABA_A_ receptors present (cf. Figure 9A). (B) Sample spike raster plots of the same TC-heterogeneous network and the same simulation conditions as that in Figure 9C, except for the addition of a GABA_A_ conductance with maximal amplitude 2 times that of the maximal GABA_B_ conductance. (C) Distributions of oscillation measures over the same 24 randomly ordered, TC-heterogeneous 200-cell networks as in Figure 9E (* p < 0.05, ** p < 0.01, *** p < 0.001, Friedman’s test for oscillation period and half activation time, repeated-measures ANOVA otherwise). The differences across simIPSCs were generally the same as in Figure 9E. (C) Figure 9E reproduced here for ease of comparison.

9) Based on the fitting of dynamic-clamp data to the model, the authors' predict high density of T-channel in dendrites and A-channel density in soma. While the rest of the work, especially the computational modeling, has focused on T-channel, the changes to A-channel have not been examined. Similar to the findings in Figure 5-6 on T-channel, it would be helpful to examine the effect of A-channel (this last point is optional to addressed in a revised manuscript).

Prior literature (Pape et al., 1994; Amarillo et al., 2014) has already examined the effects of A channels on low-threshold spikes, demonstrating the importance of the A current for controlling the amplitude and width, but not the initiation, of the LTS. It would be interesting to examine the effects of A channels on network oscillations, but since the effects of T channels are sufficient to explain most of our experimental findings, we feel like the investigation of A channels is beyond the scope of this work.

10) Given that the T-channel is involved in the generation of spindle activity during sleep, it would be helpful to include some discussion about the impact of the findings on sleep spindles.

As sleep spindles and absence seizures appear to rely on similar cellular- and circuit-level mechanisms, this point is reasonable to raise. However, sleep spindles have been shown to be primarily GABA_A_ receptor-dependent, and are likely not as dependent on GABA_B_ receptor-mediated signaling. Since our study focuses on GABA_B_ receptor modulation of thalamic circuits, we are not comfortable translating our findings to sleep spindles.

11) While the authors have developed an interesting measure called the T-channel open channel discrepancy, I feel that the effect could be well captured using conventional phase-plane or phase-space projections. It seems that the membrane voltage, T-channel activation and inactivation variables pass through a critical region in phase-space which is required for the LTS; the failure of this dynamics leads to the lack of LTS. Possibly 3D phase-space analysis of these variables could identify the critical region?(this point is optional to address in a revised manuscript).

In this work, we have demonstrated that the change of either membrane voltage V, T channel activation variable mT or inactivation variable hT relative to time is critical for LTS production. Notably, these derivatives are independent of each other. We also demonstrated that either dV/dt (slope of voltage) and d[log(mT2hT−mT,∞2hT,∞)]/dt (slope of open probability discrepancy) have some predictive power towards the production of an LTS. Therefore, we think that as opposed to 3D phase plane analysis with V, mT and hT, a phase plane analysis with dV/dt, d(mT)/dt and d(hT)/dt might provide an opportunity to identify a "critical region required for the LTS”. However, we also expect this critical region to be cell dependent. We acknowledge that such further analysis would be interesting, but we also feel that it would add unnecessary complexity for the general reader.

12) The section "Interplay between GABA_B_ receptors and T-type calcium channels " is extremely long and difficult to follow. I am sure some readers will follow all the details there but for many others the message may be lost. I suggest some rewriting to include summaries and highlight the main results.

We thank the reviewer for bringing this point to our attention. We hope to make this section as clear as possible. Therefore, we have shortened this section by moving less pertinent details to the Figure 5 and Figure 7—figure supplement 1. We have also rewritten several sections to simplify the concepts we address and moved some interpretation to the Discussion. Additionally, we agree that the original presentation of Figure 6 may have been somewhat overwhelming. Therefore, we have now split Figure 6 into two new figures (Figure 6 and Figure 7).

13) In the network simulations (Figure 8), for TC-homogeneous case, oscillations seem to be more synchronized in control vs GAT1 or GAT3 block conditions. Why is this so?

We looked into this curious observation and quantified a metric we call burst latency jitter. This parameter reflects the variability in burst latencies across trials (i.e., jitter = standard deviation). Burst latency jitter was significantly higher for both *sim*GAT1-Block and *sim*GAT3-Block across the 31 model neurons (Author response image 4). With higher burst latency jitter, oscillations become more desynchronized.

**Author response image 4. sa2fig4:** Burst latencies were more variable following simIPSCs corresponding to single GAT blockade across well-fitted model neurons. (A) Distributions of burst latency jitter (standard deviation of burst latencies across all trials) over the 31 well-fitted model neurons across GABA_B_ IPSC waveforms shown in Figure 2B (* p < 0.05, ** p < 0.01, *** p < 0.001, Friedman’s test). (B) Burst latency jitter averaged over all 31 model neurons, across 4 different GABA_B_ IPSC waveforms and 3 different conductance amplitude scales. Error bars denote 95% confidence intervals.

14) The way the manuscript is written, it is somewhat difficult to learn what is actually a precise mechanism behind the differences between effects of individual GAT1/3 blocking vs dual blocking. E.g., the Summary in discussion says: "…respectively, can be recapitulated by varying the GABA_B_ receptor activation waveform." but it does not provide a needed summary of the mechanisms discovered in the study.

We have rewritten the first paragraph of the Discussion to highlight the most important results of this work, including a summary of the discovered mechanism underlying the differences between single and dual GAT blockade. We have also clarified the section in the results wherein we highlight our mechanism.

15) Consequences of the GABA-_B_ conductance properties are reflected in the behavior of individual cells, and of oscillating networks of TCR and nRT (nucleus reticularis) neurons. Details of the GABA-_B_ conductances to be used were determined in prior publications. Experimental and numerical experiments were performed in thalamocortical slices, in dynamic clamp applied to single TCR neurons, and in simulations of 3-compartment TCR neurons alone, paired with an nRT cell, and in a TCR/nRT network. The main finding is that excessively large and prolonged GABA-_B_ conductances interfere with LTS generation; detailed analysis shows that the specific kinetic properties of T-channels are responsible, in that LTS requires an epoch where inward current is regenerative (increasing m) at the same time as inactivation is not too developed, and the latter in turn depends on instantaneous inactivation (h) being larger than would occur at equilibrium. Such large slow conductances develop when both transport blockers are applied together, but not separately. The authors provide some discussion of possible relations to spike-wave seizures, and the effects and contrary side-effects of certain anticonvulsants.I suspect that the mathematics involved is more general than discussed here for T-type calcium channels, and depends at heart (no pun intended) on the slowness of inactivation vs activation. An example might be the inability of a squid axon to fire at low frequencies, even in the presence of a low calcium buffer. The authors may, at their discretion, want to consider this larger issue (optional to address in a revised manuscript).

Comparing the dynamics of low-threshold spikes to those of action potentials would be interesting indeed. It is true that the sodium channel’s inactivation time constant is also much higher than its activation time constant, just like the T channel. However, there are additional considerations that make a direct comparison difficult, such as the fact that sodium channels are not inactivated at rest, and thus do not need to be recovered. Therefore, we expect that similar dynamics would be at play in terms of a lack of channel opening in response to slow voltage changes, but the post-inhibitory aspect would be lacking for sodium channels.

References:

Huntsman, M. M., Porcello, D. M., Homanics, G. E., DeLorey, T. M., & Huguenard, J. R. (1999). Reciprocal Inhibitory Connections and Network Synchrony in the Mammalian Thalamus. Science, 283(5401), 541–543. https://doi.org/10.1126/science.283.5401.541